# Lattice oxygen activation and local electric field enhancement by co-doping Fe and F in CoO nanoneedle arrays for industrial electrocatalytic water oxidation

Pengcheng Ye[1,5], Keqing Fang[1,5], Haiyan Wang [1]✉, Yahao Wang [1], Hao Huang [2]✉, Chenbin Mo [1], Jiqiang Ning[3] & Yong Hu [4]✉

Oxygen evolution reaction (OER) is critical to renewable energy conversion technologies, but the structure-activity relationships and underlying catalytic mechanisms in catalysts are not fully understood. We herein demonstrate a strategy to promote OER with simultaneously achieved lattice oxygen activation and enhanced local electric field by dual doping of cations and anions. Rough arrays of Fe and F co-doped CoO nanoneedles are constructed, and a low overpotential of 277 mV at 500 mA cm$^{-2}$ is achieved. The dually doped Fe and F could cooperatively tailor the electronic properties of CoO, leading to improved metal-oxygen covalency and stimulated lattice oxygen activation. Particularly, Fe doping induces a synergetic effect of tip enhancement and proximity effect, which effectively concentrates OH$^-$ ions, optimizes reaction energy barrier and promotes O$_2$ desorption. This work demonstrates a conceptual strategy to couple lattice oxygen and local electric field for effective electrocatalytic water oxidation.

Electrochemical water splitting coupling of anodic oxygen evolution reaction (OER) and cathodic hydrogen evolution reaction (HER), represents a promising technology for sustainable hydrogen generation to relieve energy and environmental crisis[1,2]. In comparison with the cathodic HER, the OER process suffering from more sluggish kinetics and a larger overpotential is identified as the major bottleneck of water electrolysis[3–5]. Various strategies such as single-atom engineering[6–8], defect regulation[9,10], lattice doping[11,12], built-in electric field construction[13,14], and surface modification[15,16], have been implemented to expedite the electron transfer process for OER. Nevertheless, the performance of the OER catalysts still cannot match that of HER, thus critically restricting their industrial applications. Gaining profound insight into the catalytic mechanism, and comprehending the correlation between the physicochemical structure of electrocatalysts and electrochemical behaviors, are of great significance to the development of highly reactive and durable OER catalysts.

At present, two typical kinds of recognized OER mechanisms that have a necessary connection with the redox metal cations or the lattice oxygen centers are proposed, which are named the adsorbate evolution mechanism (AEM) and the lattice oxygen oxidation mechanism (LOM), respectively[17]. However, as for the AEM, different oxygen intermediates including OH*, O*, and OOH* are involved, rendering a theoretically limiting overpotential to 370 mV[18,19]. Different from the AEM, the LOM undergoes an OH$^-$, O$_2^{2-}$, and O$_2$ formation pathway, which can expedite the direct coupling of lattice oxygen to bypass the limiting energy barrier of OOH* formation in the AEM process[20–22]. As a consequence, the LOM provides an appealing avenue to boost the

[1]Key Laboratory of the Ministry of Education for Advanced Catalysis Materials, Department of Chemistry, Zhejiang Normal University, Jinhua 321004, China. [2]Department of Microsystems, University of South-Eastern Norway, Borre 3184, Norway. [3]Department of Optical Science and Engineering, Fudan University, Shanghai 200438, China. [4]College of Chemistry and Materials Engineering, Zhejiang A&F University, Hangzhou 311300, China. [5]These authors contributed equally: Pengcheng Ye, Keqing Fang. ✉e-mail: chemwhy@zjnu.edu.cn; huanghao881015@163.com; yonghu@zafu.edu.cn

activity of OER electrocatalysts, such as perovskite[20], multimetallic alloy[23], and bimetallic (oxy)hydroxide[24]. Nevertheless, the switching from AEM to LOM is difficult to realize because the lattice oxygen activation is unfavorable in thermodynamics, and may typically requires a strong covalency of the transition metal-oxygen bond[20,25]. Hence, accurate adjustment of the electronic states of metal cations and oxygen ligands simultaneously for steering high metal-oxygen covalency is desired to heighten the intrinsic OER performance.

Importantly, the reactive rate of OER catalysts via LOM is still limited by mass transfer to the electrode surface, especially at industrial current densities (generally, >200 mA cm$^{-2}$)[26]. Numerous researches have verified that high-curvature nanomaterials can effectively promote efficiency and improve current density in different electrocatalytic reactions, such as electrocatalytic $CO_2$ reduction[27,28], alkyne semihydrogenation[29], OER[30], and HER[31,32]. It has been found that high-curvature nanostructures with a sharp-tip enhancement can introduce a large local electric field for concentrating electrolyte ions or aggregating reactants on the tip surface, which may simultaneously lower the reaction energy barrier and enable fast mass transfer to the active sites[27,32,33]. Nevertheless, the effect of tip-enhanced electric field on OER such as reaction intermediate formation and $O_2$ gas release has rarely been explored.

Taking the above considerations, a fascinating OER electrocatalyst should possess the features to regulate lattice oxygen redox chemistry for accommodating the LOM route by tailoring metal cations and oxygen ligands simultaneously, and also has abundant high-curvature sites to facilitate the mass transfer, which can cooperatively improve the electrocatalytic activity. However, how to integrate the two precisely is challenging and rarely reported. More importantly, a clear-cut explanation of the synergetic workings of lattice oxygen activation and local electric field enhancement as well as their interplay on electrocatalytic performance, has yet to be established.

In this study, we report a facile cation and anion dual doping strategy for simultaneous realization of lattice oxygen activation and local electric field enhancement to promote OER. Considering the similar size compatibility of Fe and Co, F and O, and the large electronegativity of F ($x = 3.98$)[34,35], well-architected Fe and F co-doped CoO nanoneedle arrays (Fe, F-CoO NNAs) with a rather rough surface are constructed, and a low overpotential of 169 mV at the current density of 10 mA cm$^{-2}$ is achieved, which is among the best results reported so far. Significantly, the Fe, F-CoO NNAs catalyst only requires the overpotentials of 234 mV and 277 mV to deliver large current densities of 100 and 500 mA cm$^{-2}$, respectively. Combining experimental results and theoretical calculations, it can be further confirmed that dual doping of Fe and F jointly upshifts Co d-band center ($\varepsilon_d$) and O 2$p$ band around the Fermi level ($E_f$), improves the covalency of the metal-oxygen band as well as activates lattice oxygen, thus facilitating the electron transfer and awakening the LOM pathway. Additionally, finite element simulations and density functional theory (DFT) in conjunction with in situ optical microscope reveal that Fe doping significantly increases the local charge density through coupling sharp tip enhancement and proximity effect, which concentrates more OH$^-$ ions, optimizes the reaction energy barrier of LOM pathway, and promotes $O_2$ bubble release. Correspondingly, the thermodynamics and kinetics barriers for OER are reduced simultaneously, leading to boosted electrocatalytic performance. These results endow the multi-scale regulation strategy of integrating lattice oxygen activation and local electric field enhancement with great potential in industrial electrocatalytic water oxidation.

## Results and discussion
### Synthesis and characterization of catalysts
The preparation of Fe, F-CoO NNAs through a facile cation and anion dual doping method is schematically illustrated in Fig. 1a. Uniform

Co(OH)F NNAs were in-situ grown on Ni foam (NF) by a facile hydrothermal method, which can be transformed into Fe, F-CoO NNAs through sequential immersion in $K_3[Fe(CN)_6]$ solution and annealing procedures. Notably, Fe and F atoms could be easily substituted into the CoO lattices due to their similar atomic sizes with Co and O, respectively. To facilitate comparison, CoO NNAs, Fe-doped CoO NNAs (Fe-CoO NNAs), and F-doped CoO NNAs (F-CoO NNAs) were also prepared using the similar strategy as Fe, F-CoO NNAs.

Field emission scanning electron microscopy (FESEM) shows that the as-prepared Co(OH)F precursor displays a smooth nanoneedle structure with a base diameter of ~150 nm (Supplementary Fig. 1). After $K_3[Fe(CN)_6]$ treatment, the crystal phase and the nanoneedle structure of Co(OH)F are almost maintained (Supplementary Fig. 2). After annealing treatment, arrays of Fe and F co-doped CoO nanoneedles are constructed, which have a rather rough surface (Fig. 1b, c). This may be due to Fe doping, leading to more structural defects in Fe, F-CoO NNAs[36]. Energy dispersive spectroscopy (EDS) results confirm the successful doping of Fe and F in the Fe, F-CoO NNAs, with atomic contents of 0.6% and 11.3%, respectively (Supplementary Fig. 3). The effect of Fe doping on the rough surface is further verified by the enhanced surface roughness of Fe, F-CoO NNAs with the increase of Fe dopant (Supplementary Fig. 4). From FESEM images (Supplementary Fig. 5), the morphologies of the CoO, Fe-CoO, and F-CoO NNAs all exhibit a similar needle-shaped structure but have a smooth or rough surface, depending on whether Fe is doped or not. Powder X-ray diffraction (XRD) patterns show no impurities except for cubic phase CoO (JCPDS No. 04-003-2614), suggesting that Fe and F atoms substitute Co and O atoms, respectively (Supplementary Fig. 6). Note that the positions of the main peaks have no significant changes for all the doped samples, which may be ascribed to the similar ion radius (Fe$^{3+}$ (0.645 Å) vs. Co$^{2+}$ (0.65 Å), and F$^-$ (1.33 Å) vs. O$^{2-}$ (1.40 Å)) and the low concentration of Fe$^{3+}$. While the intensities of the peaks reduce after F doping, which is probably due to the relatively higher content of F. Four vibrational peaks are observed in the Raman spectra, which can be assigned to the $F_{2g}$ (186 cm$^{-1}$), $E_g$ (458 cm$^{-1}$) and $A_{1g}$ (656 cm$^{-1}$) modes of Co$^{2+}$−O coordination, and the $F_{2g}$ (541 cm$^{-1}$) mode of Co$^{3+}$−O band (Supplementary Fig. 7)[37]. Detailed analysis by transmission electron microscopy (TEM) also confirms the nanosized needle-shaped morphology of Fe, F-CoO NNAs with a rather rough surface, which is in agreement with the SEM images (Fig. 1d). The lattice fringes with the spacings of 0.244 nm and 0.211 nm are observed in the high-resolution TEM (HRTEM) images, which can be attributed to the (111) and (200) planes of cubic CoO, respectively (Fig. 1e). Additionally, the selected area electron diffraction (SAED) pattern only reveals the diffraction fringes of CoO, which is consistent with the result of XRD pattern (Fig. 1f). Elemental mapping images further prove the homogeneous distribution of Co, O, Fe, and F elements throughout the nanoneedle, manifesting the successful formation of Fe, F-CoO NNAs (Fig. 1g).

To investigate the electronic structure of Fe, F-CoO NNAs, X-ray photoelectron spectroscopy (XPS) was further conducted. The XPS survey spectrum confirms the presence of Co, Fe, F, and O elements in Fe, F-CoO NNAs (Supplementary Fig. 8). The Fe 2$p$ XPS spectra of Fe, F-CoO NNAs and Fe-CoO NNAs display the Fe 2$p_{1/2}$ and Fe 2$p_{3/2}$ peaks centered at 712.4 eV and 724.5 eV, respectively, verifying the existence of Fe$^{3+}$ ions (Supplementary Fig. 9a)[38]. The F 1$s$ profiles also confirm the Co−F bond in Fe, F-CoO NNAs and F-CoO NNAs (Supplementary Fig. 9b)[39]. The Co 2$p$ XPS spectra can be deconvoluted into two spin-orbit doublets at 780.5/796.3 eV of Co$^{3+}$ and 782.7/797.7 eV of Co$^{2+}$ with two shake-up satellites[40]. The existence of Co$^{3+}$ ions might be derived from Co$^{2+}$ oxidation when exposed to air[41]. Notably, the Co 2$p$ spectrum of Fe, F-CoO NNAs is negatively shifted to lower energy compared to CoO NNAs, indicating the electronic rich state (Supplementary Fig. 9c). The Co 2$p$ spectrum of Fe, F-CoO NNAs is located between Fe-CoO NNAs and F-CoO NNAs, illustrating that the Fe and F dopants might act as electronic donors and electronic acceptors,

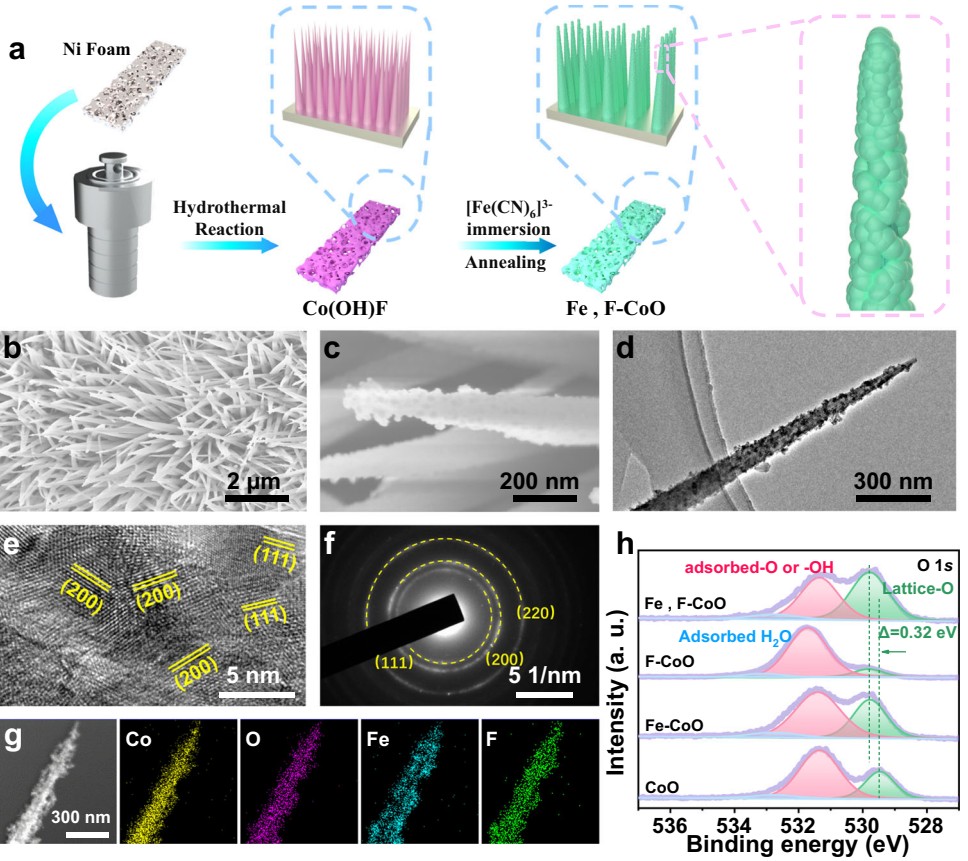

**Fig. 1 | Preparation, morphology and structure of the Fe, F-CoO NNAs catalyst.**
**a** Schematic illustration of the synthetic procedure of the Fe, F-CoO NNAs catalyst with a rough surface. **b** FESEM image. **c** Enlarged FESEM image. **d** TEM image. **e** HRTEM image. **f** SAED pattern. **g** STEM image and corresponding elemental mapping images. **h** High-resolution O 1s XPS spectra of the Fe, F-CoO NNAs, F-CoO NNAs, Fe-CoO NNAs and CoO NNAs catalysts.

respectively, and jointly change the electronic structure of CoO. The O 1s spectra show three oxygen species including lattice-O, adsorbed-O or hydroxyl (−OH), and adsorbed $H_2O$ (Fig. 1h). In particular, the peaks of lattice-O for the Fe-CoO NNAs, F-CoO NNAs, and Fe, F-CoO NNAs all reveal a positive shift relative to that of CoO NNAs, unveiling a higher electron-cloud density and higher lattice oxygen activity in the doped CoO samples[42]. Moreover, among the four samples, the largest content of lattice oxygen is obtained in Fe, F-CoO NNAs, which may benefit the OER process (Supplementary Table 1)[43].

The electronic and chemical coordination structures of the catalysts were further explored using X-ray absorption spectroscopy (XAS). Figure 2a presents the O K-edge spectra of the CoO NNAs, F-CoO NNAs and Fe, F-CoO NNAs, in which the O K-edge pre-edge peak centered at 532 eV is attributed to the hybridization O 2p-state with the Co 3d. As described, the O K-edge pre-peak almost disappears for F-CoO NNAs, suggesting that oxygen release occurs with F doping. It is notable that compared with CoO NNAs, Fe, F-CoO NNAs present a more intense pre-edge peak, suggesting the strengthened metal-oxygen covalency[44,45]. This result is consistent with those from O 1s XPS spectra. The Co K-edge X-ray absorption near edge structure (XANES) spectra show that the absorption edges of the three samples are located between standard CoO and $Co_2O_3$, manifesting the valence state of Co between +2 and +3 (Fig. 2b). A shift to lower energies is observed for Fe, F-CoO NNAs compared to CoO NNAs, suggesting the lower Co valence. The Fourier-transformed extended X-ray absorption fine structure (FT-EXAFS) spectra of Co K-edge reveal two shells at ~1.4 and ~2.4 Å, corresponding to the Co−O and Co−Co scattering paths, respectively (Fig. 2c). The almost overlapped Co K-edge oscillation plots and the similar distances of Co−O and Co−Co in un-doped and doped samples indicate that the incorporation of Fe and F does not

cause significant structural changes in CoO (Supplementary Figs. 10, 11, Supplementary Table 2). This is further confirmed by the similar wavelet transform (WT) plots (Fig. 2d−i), agreeing well with the observations from XRD and Raman.

## Electrocatalytic OER performance

The OER activity of Fe, F-CoO NNAs was evaluated in 1 M KOH electrolytes using a conventional three-electrode system, with CoO NNAs, F-CoO NNAs, and Fe-CoO NNAs for comparison. Linear sweep voltammetry (LSV) curves evidence the best OER performance of the Fe, F-CoO NNAs with the lowest overpotential among all the samples (Fig. 3a, Supplementary Fig. 12). As shown, the Fe, F-CoO NNAs only require an overpotential of 169 mV at the evaluation metrics of 10 mA cm$^{-2}$, surpassing those of F-CoO NNAs (213 mV), Fe-CoO NNAs (220 mV), CoO NNAs (304 mV), commercial $IrO_2$ (294 mV) (Fig. 3b). Notably, to deliver large current densities of 100 mA cm$^{-2}$ and 500 mA cm$^{-2}$, extremely low overpotentials of 234 mV and 277 mV are achieved for the Fe, F-CoO NNAs, respectively, which are superior to those of the comparisons. Also, the distinguished OER performance of Fe, F-CoO NNAs is revealed by the lower Tafel slope of 41.6 mV dec$^{-1}$ compared to those of F-CoO NNAs (55.8 mV dec$^{-1}$), Fe-CoO NNAs (60.1 mV dec$^{-1}$), and CoO NNAs (74.7 mV dec$^{-1}$), implying the faster OER kinetics (Fig. 3c). The corresponding Tafel slopes and over-potentials are comparable to those of the state-of-the-art transition metal-based OER electrocatalysts (Fig. 3d & Supplementary Tables 3,4). Besides, the real-time potential of Fe, F-CoO NNAs hardly increases during the continuous testing for 300 h at large current densities of 100 mA cm$^{-2}$ and 500 mA cm$^{-2}$ (Fig. 3e). This result is also comparable or even superior to most transition metal-based OER electrocatalysts (Supplementary Table 5). The robust durability is also

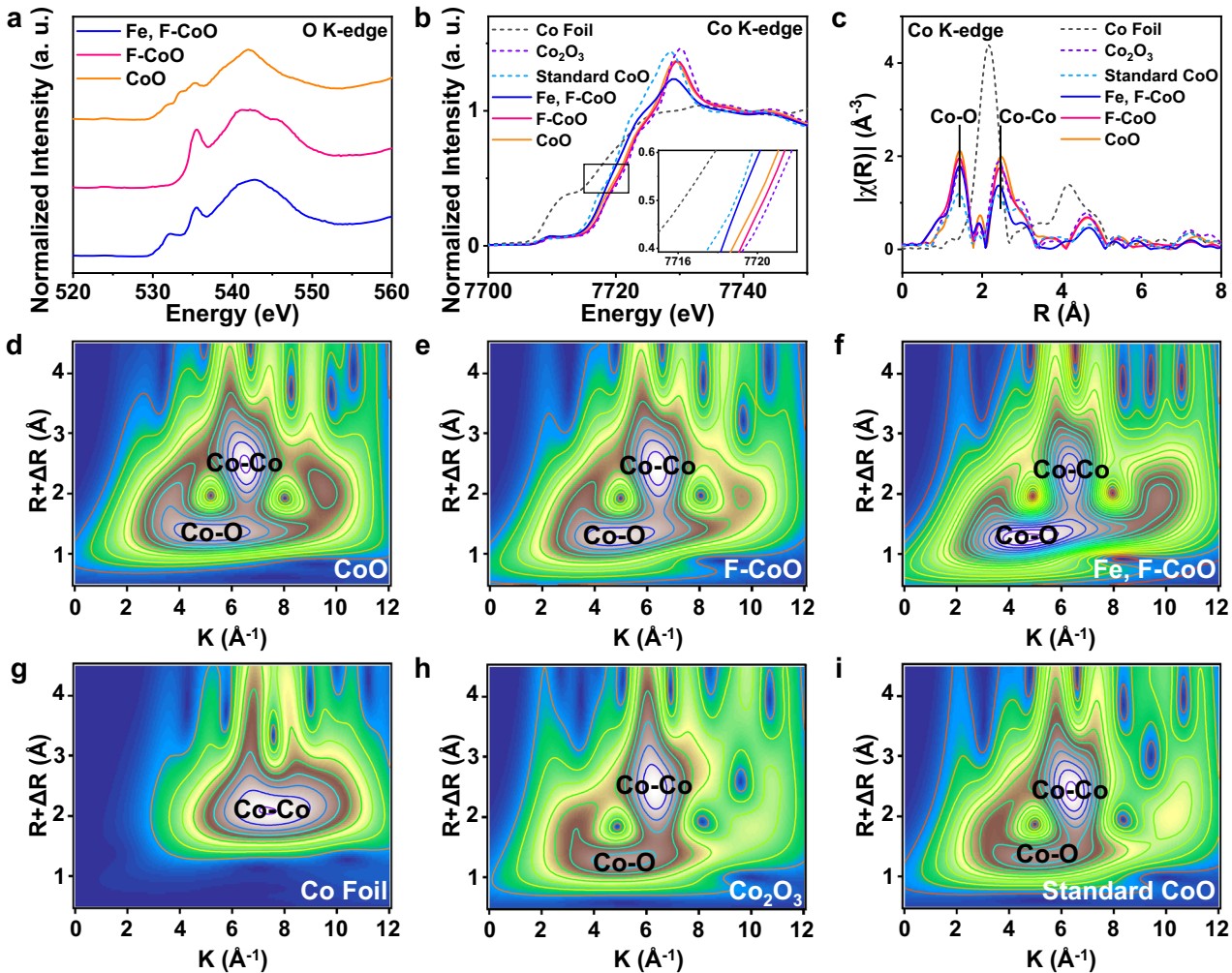

**Fig. 2 | Electronic and chemical coordination structures of the Fe, F-CoO NNAs and references. a** Normalized O K-edge spectra of CoO NNAs, F-CoO NNAs, and Fe, F-CoO NNAs. **b** Normalized Co K-edge XANES spectra. **c** FT-EXAFS spectra of Co K-edge. **d–i** WT-EXAFS analysis of CoO NNAs, F-CoO NNAs, Fe, F-CoO NNAs, and the standard references including Co foil, standard CoO and $Co_2O_3$.

verified with the LSV curves (Supplementary Fig. 13). FESEM and XRD further demonstrate that the morphology and crystal structure show negligible changes after the cycling test, suggesting their robust stability and great potential in practical applications (Supplementary Fig. 14).

Furthermore, the intrinsic activities of the as-prepared electrocatalysts were evaluated by normalizing the LSV curves with the electrochemical active surface area (ECSA) to exclude the effect of geometric structure. Fe, F-CoO NNAs obtain the largest electrical double-layer capacitance ($C_{dl}$) of 6.46 mF $cm^{-2}$, corresponding to an ECSA of 81 $cm^2$ (Supplementary Figs. 15, 16). Besides, in contrast to F-CoO NNAs, Fe-CoO NNAs, and CoO NNAs, the Fe, F-CoO NNAs exhibit a more favorable intrinsic activity as demonstrated by the higher ECSA-normalized current densities (Supplementary Fig. 17). In particular, for Fe, F-CoO NNAs, the ECSA-normalized current density can reach 4.41 mA $cm^{-2}$ at an overpotential of 300 mV, which is about 1.9 times of F-CoO NNAs (2.37 mA $cm^{-2}$), and even 6.7 times and 17.6 times greater than Fe-CoO NNAs (0.66 mA $cm^{-2}$) and CoO NNAs (0.25 mA $cm^{-2}$), respectively (Fig. 3f). The above results further verify that the intrinsic electrocatalytic activity of CoO is boosted by Fe, F dual doping.

## Kinetic study

Afterward, the kinetic processes, with regard to the electron transport and the adsorption/desorption of reactants on the electrode surface,

were studied by electrochemical impedance spectroscopy (EIS). Nyquist plots manifest that the charge transfer resistance ($R_{ct}$) of Fe, F-CoO NNAs is significantly smaller than those of F-CoO NNAs, Fe-CoO NNAs, and CoO NNAs, indicating the faster charge transfer rate and accelerated electrocatalytic kinetics (Supplementary Fig. 18, Supplementary Table 6)[46]. Operando EIS was carried out to better expound the reaction kinetics during the oxygen evolution process. As depicted, the electronic transfer and the adsorption of reactants processes are markedly facilitated with the increase of voltage, contributing to faster OER kinetics (Fig. 3g and Supplementary Fig. 19)[47]. Besides, the relatively small phase angles in the low-frequency region of Bode phase plots represent rapid electron transport inside the electrodes, while the swiftly descending of the phase angles in the high-frequency regions implies the accelerated interface reaction during the electrocatalytic process (Supplementary Fig. 20)[48]. Fe, F-CoO NNAs exhibit lower phase angles in comparison with those of F-CoO NNAs, Fe-CoO NNAs and CoO NNAs, confirming a more active interface (Fig. 3h). From the above results, we can deduce that Fe, F dual doping effectively enhance the intrinsic OER activity of CoO and promote the interfacial charge/electron transfer at the electrode interface, thus promoting the OER process.

## Exploration of reaction mechanism

Given the high OER activity of the Fe, F-CoO NNAs, the pH dependence on OER activities was also investigated, which can provide insights into

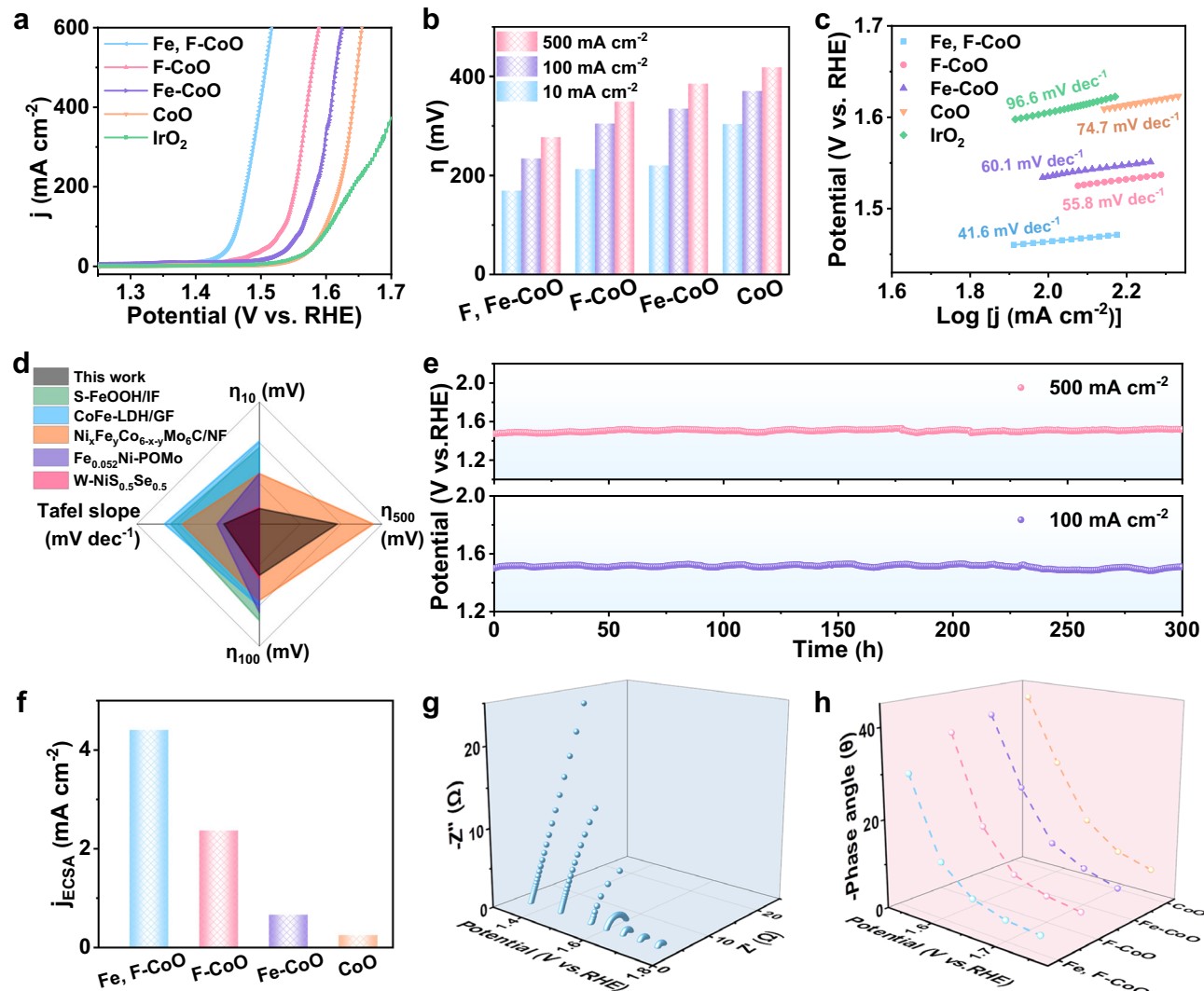

**Fig. 3 | OER performances of the Fe, F-CoO NNAs, F-CoO NNAs, Fe-CoO NNAs, CoO NNAs, and commercial IrO₂ catalysts (loading: 4.5 mg cm⁻²) in 1 M KOH.** **a** LSV curves with 90% IR compensation. **b** Corresponding overpotentials at 10, 100 and 500 mA cm⁻². **c** Tafel plots. **d** Comparison of the Tafel slopes and overpotentials at 10, 100, and 500 mA cm⁻² with the state-of-the-art transition metal-based OER electrocatalysts. **e** Chronopotentiometry measurements of the Fe, F-CoO NNAs at 100 and 500 mA cm⁻². **f** Specific activities by normalizing the current densities with ECSA at an overpotential of 300 mV. **g** Operando EIS spectra of Fe, F-CoO NNAs at various potentials. **h** Phase angles of the peaks at 1.55–1.75 V vs. RHE.

the reaction mechanism. As profiled in Supplementary Fig. 21, the current densities of the Fe, F-CoO NNAs increase steeply with the increase of pH from 12.5 to 14.0, suggesting a nonconcerted proton-electron transfer pathway[21]. The pH dependence is closely related to the proton reaction order ($\rho$), namely, Fe, F-CoO NNAs (0.98) > F-CoO NNAs (0.95) > Fe-CoO NNAs (0.80) > CoO NNAs (0.65) (Fig. 4a). Generally, a $\rho$ value close to 1 manifests a preferential LOM, and therefore Fe, F-CoO NNAs and F-CoO NNAs may undergo the LOM pathway[49]. Besides, the strongest pH-dependent behavior of Fe, F-CoO NNAs evidences that the lattice oxygen redox chemistry of CoO can be activated most effectively through Fe and F dual substitution[24]. To verify the LOM mechanism, tetramethylammonium cation (TMA⁺) is adopted as a chemical probe of negatively charged peroxo/superoxo-like ($O_2^{2-}/O_2^-$) intermediates due to their strong electrostatic interactions (Fig. 4b)[21,24]. Distinctly, the current density of Fe, F-CoO NNAs declines significantly in 1 M TMAOH with the largest reduction factor of 90.7% at 1.55 V among the samples, reflecting that the LOM process in Fe, F-CoO NNAs is severely hindered by TMA⁺ (Fig. 4c). In contrast, Fe-CoO NNAs and CoO NNAs both show incredibly tiny fractional variations in OER activity and kinetics, indicting the preferred AEM process in these two catalysts (Supplementary Fig. 22). These results

were directly demonstrated with Raman spectra, in which the typical peaks of TMA⁺ at 753 cm⁻¹ and 953 cm⁻¹ are observed for Fe, F-CoO NNAs and F-CoO NNAs, but absent for Fe-CoO NNAs and CoO NNAs (Supplementary Fig. 23)[24]. The above analyses prove that the OER mechanism switches from AEM to LOM after F doping, and more importantly, Fe and F dual doping is more favorable for the LOM pathway in contrast with other single-doped CoO catalysts. Furthermore, in situ surface-enhanced Raman spectroscopy was carried out to gain a deeper insight into the active sites of Fe, F-CoO NNAs. As shown in Fig. 4d, three spectral features of CoO, assigned to the $E_g$ vibration mode of Co (II) − O, $F_{2g}$ mode of Co (III) − O band, and $A_{1g}$ of Co (II) − O band, are observed at around 473, 544, and 691 cm⁻¹ at 1.23 V[37,50]. Notably, when the potential is elevated to 1.33 V, the two peaks of Co (II) − O coordination at 473 and 691 cm⁻¹ attenuate slightly, while the intensity of Co (III) − O peak increases significantly. And this phenomenon is more evident with the further increased potential, suggesting more Co (II) is transformed into Co (III) species. This may be because the low-valence metal sites can be spontaneously and rapidly oxidized by alkaline electrolytes, which is believed favorable for activating lattice oxygen[43,51]. Moreover, a broad feature in the range of 900 - 1200 cm⁻¹ gradually appears in the surface-enhanced Raman

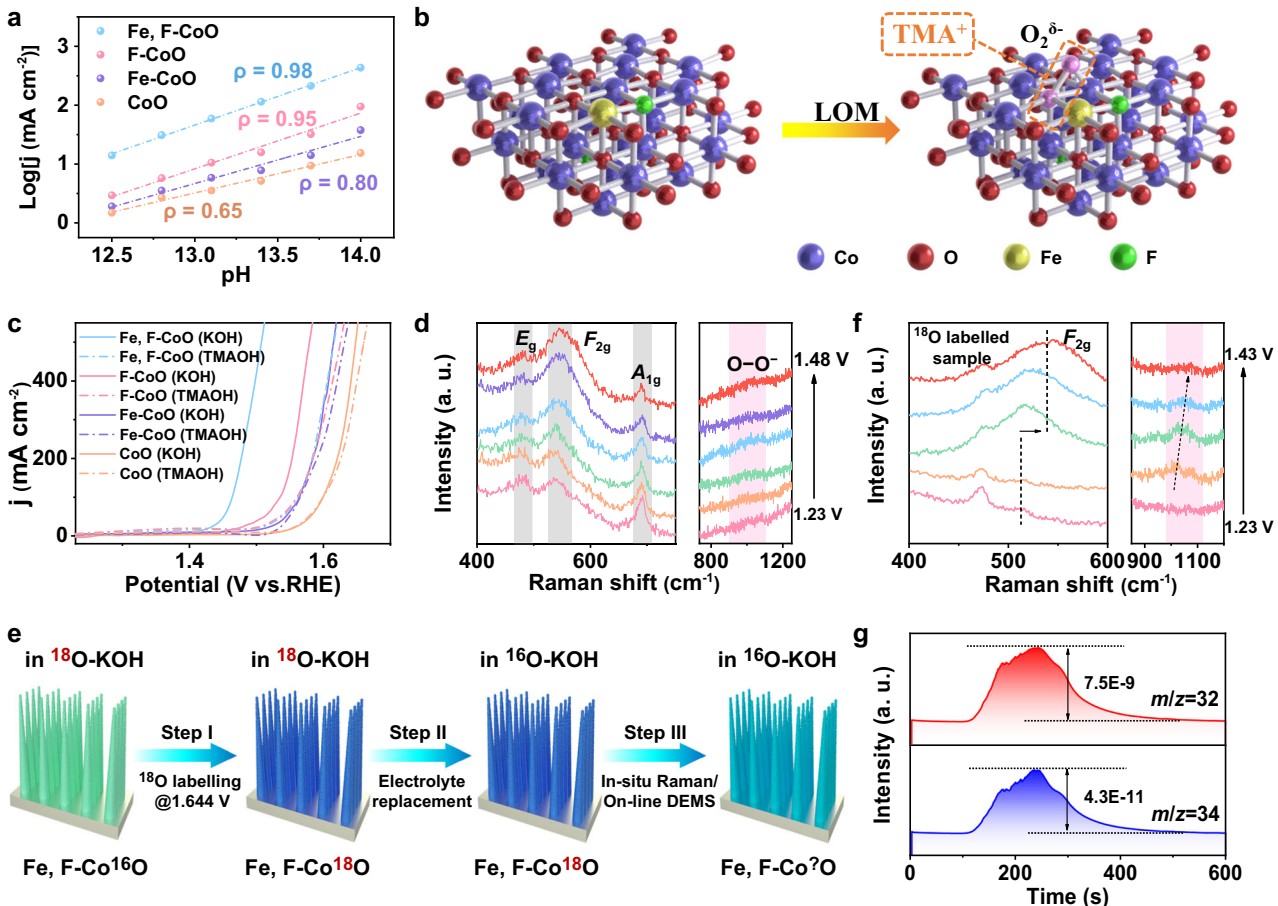

**Fig. 4 | Evidence for the LOM pathway and identification of active sites.**
**a** Current density at 1.55 V vs. RHE as a function of pH value of the Fe, F-CoO NNAs, F-CoO NNAs, Fe-CoO NNAs, and CoO NNAs catalysts in KOH electrolytes (pH: 12.5 - 14.0). **b** Schematic illustration of the chemical recognition of negatively charged peroxo/superoxo-like species in LOM using TMA$^+$ as a chemical probe. **c** LSV profiles of the Fe, F-CoO NNAs, F-CoO NNAs, Fe-CoO NNAs, and CoO NNAs catalysts in 1 M KOH and 1 M TMAOH with 90% IR compensation. **d** In situ Raman spectra of Fe, F-CoO NNAs in 1 M KOH from 1.23 to 1.48 V vs. RHE. **e** Schematic illustration of the oxygen isotope labeling experiments. **f** In situ Raman spectra of Fe, F-Co$^{18}$O NNAs. **g** On-line DEMS spectra of Fe, F-Co$^{18}$O NNAs.

spectra at 1.33 V and above, with a zenith at around ~1020 cm$^{-1}$, which is assigned to a surface superoxol-like species (OO$^-$) that are often recognized as the active oxygen species of LOM[24,43,52].

Due to the isotope effect, the LOM pathway was further identified by coupling oxygen isotope with electrochemical in situ Raman spectroscopy. A three-step strategy is adopted for the oxygen isotope labeling experiments (Fig. 4e)[53]. First, the as-prepared Fe, F-CoO NNAs are labeled in 0.1 M KOH H$_2^{18}$O aqueous solution (denoted as $^{18}$O-KOH) with a chronoamperometry method for 20 min at 1.664 V versus RHE. As indicated in Supplementary Fig. 24, the $F_{2g}$ and $A_{1g}$ peaks of the Co−O band both shift to lower frequencies, indicating the successful labeling of lattice oxygen with $^{18}$O in Fe, F-CoO NNAs[25,53]. Subsequently, the $^{18}$O-labeled Fe, F-CoO NNAs (denoted as Fe, F-Co$^{18}$O NNAs) sample was transferred into 0.1 M KOH-H$_2^{16}$O ($^{16}$O-KOH) aqueous solution and finally, Raman spectra were in situ collected in $^{16}$O-KOH electrolytes. Raman spectra reveal that the peaks of $F_{2g}$ mode of Co−O band and OO$^-$ band gradually moved toward higher frequencies with the elevated potentials, demonstrating that the lattice oxygen of Fe, F-CoO NNAs can exchange with the electrolyte and get involved in the OER process (Fig. 4f). Furthermore, $^{18}$O isotope technology combined with on-line differential electrochemical mass spectrometry (DEMS) was used to detect O$_2$ gas generated during the OER. As shown in Fig. 4g, the signals for mass-to-charge ratio ($m/z$) of 32 and 34, corresponding to $^{32}$O$_2$ ($^{16}$O$^{16}$O) and $^{34}$O$_2$ ($^{16}$O$^{18}$O), respectively, are detected for Fe, F-Co$^{18}$O NNAs, confirming the involvement of lattice oxygen

in the OER. These observations provide more compelling evidence for the LOM mechanism.

## Theoretical insights into reaction mechanism

To elucidate the origins of the different electrochemical behaviors of doped CoO, and also clarify the impacts of Fe and F dual doping on electronic structure and reaction mechanism, theoretical studies were elaborately performed based on DFT calculations. The CoO (200) slabs with Fe and/or F atoms situated on the top sites were chosen as the computational models for three reasons (Supplementary Fig. 25). First, the (200) atomic layer is stable and usually chosen for theoretical calculations in most works. Second, HRTEM and XRD reveal that the CoO (200) are the exposed active surfaces. Third, the segregation energy ($E_{seg}$) for Fe and F dopants in the CoO (200) surface, which is defined as the total energy difference of the CoO slab with dopant in the first layer and sub-layer (bulk), are −1.94 eV and 0.79 eV, respectively (Supplementary Fig. 26)[54]. This result indicates that Fe prefers to dope in the surface and F can dope in the surface and bulk, consistent well with the XPS and EDS mapping results. Moreover, it is believed that electrochemical reactions often occur at the oxide surface and only the surface allows the formation of O−O bonds because of the low flexibility[55]. Therefore, the CoO models with Fe and F atoms situated on the top site were constructed. As illustrated in the 3$d$ orbital partial density of states (PDOS) of the Co atom (the active site), compared with pure CoO slab, Fe doping can apparently upshift the $\varepsilon_d$ of the Co atom approaching the $E_f$, while the F atom has little impact on the Co

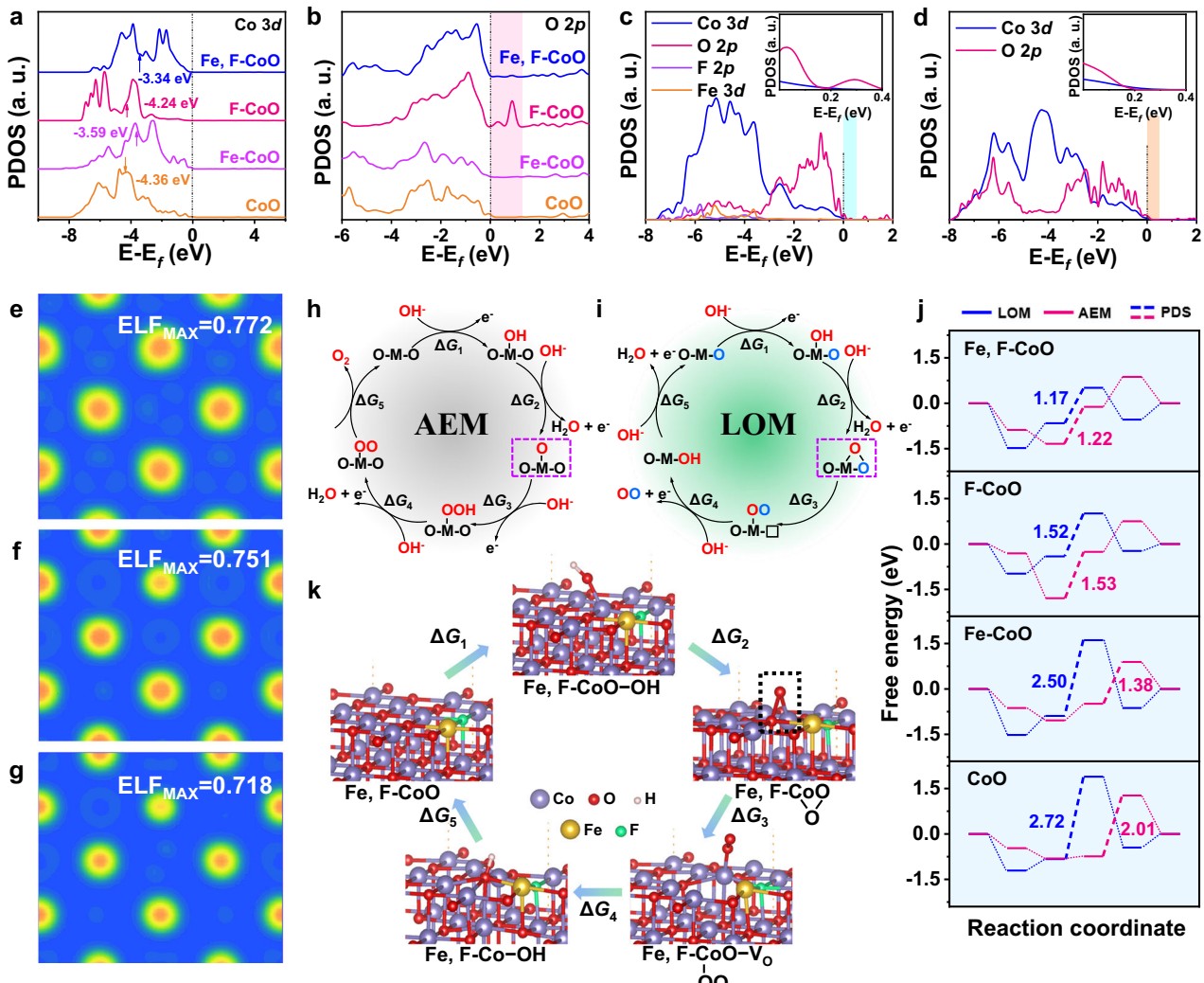

**Fig. 5 | DFT calculations. a** Calculated PDOSs of the Co 3*d* orbits (active site). **b** Calculated PDOSs of O 2*p* orbits (active site). **c** Co 3*d*, O 2*p*, F 2*p*, and Fe 3*d* PDOS spectra of the Fe, F-CoO slab with an inset of the enlarged shaded area. **d** Co 3*d* and O 2*p* PDOS spectra of the CoO slab with an inset of the enlarged shaded area. ELF diagrams of (**e**) Fe, F-CoO, (**f**) F-CoO, and (**g**) CoO slabs. Schematic illustrations of the OER mechanisms: (**h**) AEM and (**i**) LOM. **j** Gibbs free energy diagrams for OER on the Fe, F-CoO, F-CoO, Fe-CoO, and CoO slabs via AEM and LOM pathways. **k** Models of the different intermediates in the LOM pathway on the Fe, F-CoO slab.

3*d* orbital (Fig. 5a). More importantly, the Co atom in Fe, F-CoO slab obtains the highest $\varepsilon_d$ of −3.34 eV among the four models, which effectively strengthens the interaction between electrocatalysts and oxygen intermediates owing to the synergistic effect of Fe and F heteroatoms[56]. From the 2*p* orbital PDOS of the active O atom, unoccupied oxygen states (shaded portion) above the $E_f$ are observed after the incorporation of F, allowing more electrons of oxygen intermediates to enter (Fig. 5b). It should be noted that the single Fe dopant has negligible influence on the increased O 2*p* states around the $E_f$. Compared to the other doped CoO slabs, the higher energy level of the O 2*p* band for Fe, F co-doped slab, closer to $E_f$, indicates a higher oxygen ion mobility, which may favor the oxygen vacancy formation and OH− absorption[57]. The synergistic effect of Fe and F atoms on the electronic structure can also be verified from the calculated PDOS of Co 3*d*, O 2*p*, and Fe 3*d* orbits. As depicted in Fig. 5c, substantial overlaps are found in the Co 3*d*, O 2*p*, and Fe 3*d* orbits of Fe, F-CoO, suggesting a strong interaction between Co and Fe atoms as well as a covalent hybridization between Co sites and the oxygen ligands. Additionally, the introduction of Fe and F causes more unoccupied metal-oxygen bands to enter above $E_f$ (shaded portion), indicating a stronger metal-oxygen covalency in Fe, F-CoO (Fig. 5c and d)[43]. The covalent characteristic can also be quantitatively studied by electronic

localization function (ELF) analysis, in which the electronic locality is reduced after Fe, F dual substitution, indicating a strengthened covalent character of metal-oxygen bond (Fig. 5e–g)[58]. In addition, non-bonding oxygen states are usually recognized as crucial for lattice oxygen oxidation. For CoO and Fe-CoO, there is one kind of oxygen: O1, O-M-O (M=Co, Fe). For F-CoO and Fe, F-CoO, there are two kinds of oxygen: O1, O-M-O and O2, O-M-F (M=Co, Fe). As shown in Supplementary Fig. 27, large amounts of high-energy non-bonding oxygen states are generated, which are associated with the O2 bonding environment. High metal-oxygen covalency and the existence of non-bonding oxygen states both are beneficial for triggering lattice oxygen to participate in the OER process and promoting the facile electron transfer between metal and oxygen intermediates ($O_2^{2-}/O^{2-}$), thereby enhancing the OER kinetics[24,43,59].

The oxygen evolution processes on the different configurations and the Gibbs free energy for the evolution of different intermediates in both the AEM and the LOM pathways were further carried out. The AEM and LOM routes with five different elementary steps are schematically illustrated in Fig. 5h, i. As shown, the AEM pathway experiences concerted proton-electron transfer processes with four reactive oxygen species, namely, M − OH, M − O, M − OOH, and M − OO (M represents the transition metal)[24]. Different from the

AEM, the LOM undergoes the route of $O-M-OH$, $MOO$ (dashed box), $OO-M-\Box$, and $M-OH$ (bold $O$ is lattice oxygen active site, $\Box$ denotes oxygen vacancy), which involves the direct coupling of *O intermediate with active lattice oxygen. The free energy diagrams reveal that the OER mechanisms of the CoO and Fe-CoO slabs both follow a conventional AEM route, since the potential determining step (PDS) in an AEM route, that is Co−OO formation, holds lower energy compared to oxygen vacancy formation in a LOM pathway (Fig. 5j & Supplementary Figs. 28, 29). Additionally, the energy barrier of the PDS for the AEM pathway decreases from 2.01 to 1.38 eV after Fe doping (Supplementary Table 7). This is in great alignment with the result that Fe-doping causes a positive shift of $\varepsilon_d$ of active Co sites, thus enhancing the OER process. However, once F is doped into CoO, as seen from F-CoO and Fe, F-CoO slabs, the free energy change for the formation of oxygen vacancy in the LOM pathway decreases dramatically, implying that F doping plays a critical role in stabilizing the oxygen vacancy (Fig. 5k & Supplementary Figs. 30, 31). As a result, the LOM pathway is thermodynamically more favorable for F-CoO and Fe, F-CoO slabs with a lower energy barrier compared to the AEM pathway, further verifying the effective activation of lattice oxygen in F-CoO and Fe, F-CoO slabs. This is also evidenced by the prominent upshift of O 2p bands toward $E_f$, as observed in the O 2p PDOS diagrams (Supplementary Fig. 32). What's more, the Fe, F-CoO slab delivers the lowest energy uphill of 1.17 eV to complete the OER loop among the four samples, thereby exhibiting supreme intrinsic activity and a decreased overpotential for OER. These are in good alignment with the optimized electron structures of Co and O atoms, and the highest metal-oxygen covalency in the Fe, F-CoO slab as demonstrated above.

### Effects of local electric field

Furthermore, the pivotal role of Fe doping-induced rough nanoneedle structure for boosted electrocatalytic activity was investigated. Finite element method simulations were first employed to explore the distributions of charges and $OH^-$ ions close to the electrode surface. Four models with different radii and disparate surfaces were constructed to simulate the immersion of CoO electrodes in 1.0 M KOH electrolyte (Supplementary Fig. 33). Evidently, positive charge density is mainly centralized at the tips of nanoneedles and displays a 24-fold increase with the sharpening of the top radius from 110 to 3.2 nm (Fig. 6a, b). The tip-enhanced local current is probably due to the electron migration resulting from the electrostatic repulsion effect[27]. To simulate the rough surface, 1.5 nm spherules were decorated on the nanoneedle surface as a model, in which each spherule can serve as a tiny tip for charge accumulation. Intriguingly, a larger positive charge density of 0.094 C m$^{-2}$ is found, approximately 1.4 times that of the smooth one (Fig. 6c). Moreover, decreasing the distance between the spherules from 16 to 6 nm is favorable to converging the positive charge density surrounding the nanoneedle surface and increasing the charge density dramatically, suggesting a conspicuous proximity enhancement (Fig. 6d). The charge density can reach 0.11 C m$^{-2}$, corresponding to an electric field of $1.58 \times 10^{-2}$ V/Å. To quantitatively reckon the influence of the electric field on the distribution of surface $OH^-$ ions, a Gouy-Chapman-Stern model is then adopted for portraying the concentration of $OH^-$ in the Helmholtz layer of the electrical double layer near the electrode surface. As depicted, due to the coupling of tip-enhanced electric field and the adjacency effect, the $OH^-$ concentration on rough needle-like structure with small spherule gaps (6 nm) reaches the highest, increased by 170 % and 20 % compared to smooth needle tips and the model with large spherule gaps (16 nm), respectively (Fig. 6e−h). A higher $OH^-$ concentration means a larger pH, which thermodynamically favors the OER[30]. Furthermore, the effect of local electric field on the OER process was investigated by DFT. As illustrated in Supplementary Fig. 34, at an electric field of $1.58 \times 10^{-2}$ V/Å, the energy uphill of PDS for LOM on the Fe, F-CoO slab is lowered to

0.99 eV, indicating the accelerated lattice oxygen activation kinetics. Meanwhile, the release rate of $O_2$ bubbles on the rough nanoneedle structure is significantly faster than that on flat surfaces due to the smaller stretch force, as demonstrated by the in situ optical microscope images (Fig. 6i, j & Supplementary Movies 1,2)[60,61]. Hence, the Fe, F-CoO NNAs can induce a stronger local electric field to enrich $OH^-$ ions at the electrode surface, expose more active sites to $OH^-$ ions, optimize the energy barrier of LOM, as well as promote $O_2$ release, thus promoting the alkaline OER process.

### Industrial overall water splitting performance

The applicability of the Fe, F-CoO NNAs catalyst in industrial overall water splitting was also investigated. The configuration of the alkaline two-electrode electrolyzer is schematically presented in Supplementary Fig. 35, employing Fe, F-CoO NNAs and Pt/C catalysts as the anode and cathode, respectively. Polarization curves manifest that the Fe, F-CoO NNAs only require 1.41 V to deliver in 1.0 M KOH electrolyte at the current density of 10 mA cm$^{-2}$, which is much lower than those of F-CoO NNAs, Fe-CoO NNAs, CoO NNAs, and $IrO_2$ catalysts (Supplementary Fig. 36). Especially, low cell voltages of 1.57 V and 1.70 V are achieved for the as-assembled Fe, F-CoO NNAs (+) || Pt/C (−) system, to reach the current densities of 100 mA cm$^{-2}$ and 500 mA cm$^{-2}$. These values also surpass most of the state-of-the-art electrocatalysts (Supplementary Table 8). More importantly, the voltage of the alkaline electrolyzer can be maintained for 60 h without obvious loss at industrial current densities from 100 to 500 mA cm$^{-2}$ (Supplementary Fig. 37). Thus, the Fe, F-CoO NNAs show great potential in large-scale water splitting.

In summary, this work demonstrates the importance of cation and anion dual doping on multi-scale regulation of lattice oxygen redox chemistry and local electric field for enhancing OER activity, through a combination of experiments, XAS, in situ Raman spectroscopy, $^{18}O$ isotope technology, DEMS, DFT calculations, finite element method simulations, and in situ optical microscope. Rough Fe, F-CoO NNAs with triggered lattice oxygen sites and an enhanced local electric field have been successfully constructed, which boost the OER process. Studies show that relative to single doping, the dually doping of Fe and F into CoO introduces stronger metal-oxygen covalency and higher lattice oxygen reactivity by regulating the Co $\varepsilon_d$ and the O 2p band simultaneously. In particular, Fe and F dopants mainly adjust the electronic states of the Co atom and oxygen ligand, respectively, resulting in disparate electrochemical behaviors and OER mechanisms in Fe-CoO NNAs and F-CoO NNAs. Moreover, it has been proven that Fe doping induces a sharp tip enhancement and a proximity effect, and contributes to a stronger electric field, which favors the enrichment of $OH^-$ around the active sites, optimizes reaction energy barrier, as well as promotes the release of $O_2$ bubbles. As a result, the as-prepared Fe, F-CoO NNAs exhibit remarkably low overpotentials of 169 mV at the current density of 10 mA cm$^{-2}$ and 277 mV at the industrial current density of 500 mA cm$^{-2}$ with robust durability, which surpass most of the state-of-the-art catalysts. These results provide new insights into the integration of lattice oxygen redox chemistry and local electric field, which may open a new path to design high-efficiency catalysts for accelerating the OER kinetics, and also deepen the understanding of the structure-activity relationship and electrocatalytic mechanism. More broadly, this work could offer some promising views for other catalytic reactions as well as energy storage systems that involve oxygen chemistry or mass transfer kinetics.

## Methods

### Chemicals

Co(NO$_3$)$_2$·6H$_2$O was purchased from Shanghai Aladdin Chemical Reagents Co., Ltd. Other chemicals such as urea, NH$_4$F, K$_3$[Fe(CN)$_6$],

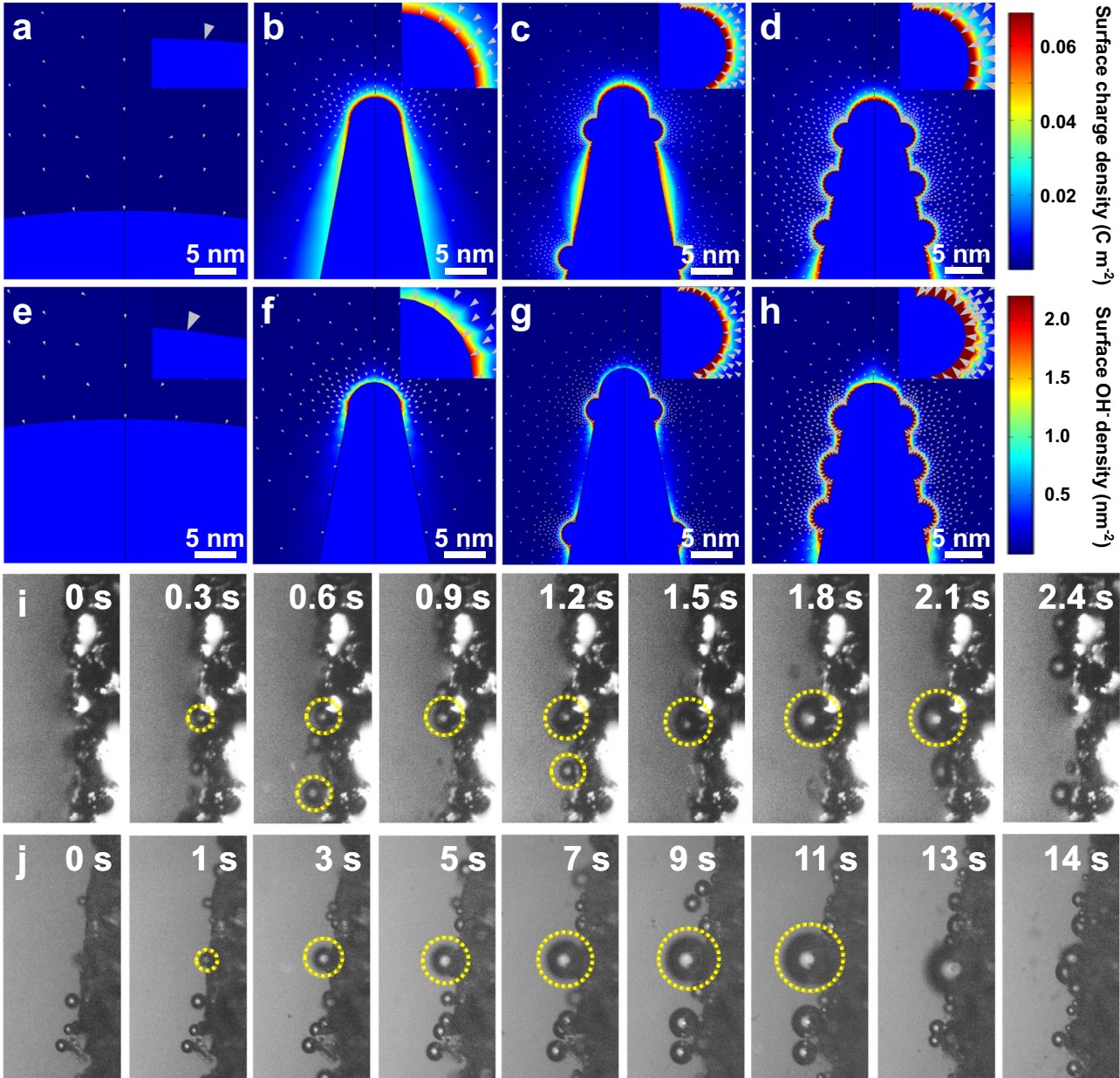

**Fig. 6 | Finite element simulations, in situ optical microscope pictures.** Enlarged images of (**a–d**) surface positive charge density distribution and (**e–h**) surface OH⁻ density distribution on the tips of different electrode models: (**a, e**) The nanoneedles with a top radius of 110 nm and a bottom radius 150 nm, (**b, f**) The smooth nanoneedles with a top radius of 3.2 nm and a bottom radius of 60 nm, The nanoneedles (top radius: 3.2 nm and bottom radius: 60 nm) with 1.5 nm spherules on the surface, and the gaps between the spherules are (**c, g**) 16 nm and (**d, h**) 6 nm. In situ optical microscope pictures of (**i**) Fe, F-CoO NNAs and (**j**) CoO NNAs in 1 M KOH electrolytes.

and KOH were obtained from Sinopharm Reagent Co., Ltd. All the reagents were directly used without further purification.

### Synthesis of Co(OH)F NNAs

Typically, NF was ultrasonically cleaned in 3 M HCl solution for 30 min, then rinsed with de-ionized (DI) water and anhydrous ethanol three times, respectively. The cleaned NF was dried in an oven at 60 °C overnight before use. Co(OH)F NNAs were grown on NF via a simple hydrothermal method. Specifically, a mixed solution of 0.125 M $Co(NO_3)_2\cdot6H_2O$, 0.625 M urea, and 0.125 M $NH_4F$ was transferred into a 25 mL Teflon-lined stainless steel autoclave containing pretreated NF. The autoclave was maintained at 120 °C for 6 h. After cooling to room temperature, the NF was removed and rinsed with DI water and anhydrous ethanol sequentially. Co(OH)F NNAs on NF were finally obtained after drying in an oven at 60 °C for 12 h.

### Synthesis of Fe, F-CoO NNAs

In a typical synthesis, the as-prepared Co(OH)F NNAs supported on NF were immersed in 0.01 M $K_3[Fe(CN)_6]$ solution at 50 °C for 20 min. Then, the product was washed with DI water and anhydrous ethanol by ultrasonic and dried at 60 °C. Subsequently, the as-obtained product was annealed at 300 °C for 4 h with a heating rate of 2 °C $min^{-1}$ to obtain Fe, F-CoO NNAs. The loading of the catalyst was about 4.5 mg $cm^{-2}$. Fe, F-CoO NNAs with different Fe contents can be prepared by changing the concentrations of $K_3[Fe(CN)_6]$ (0.005 M, 0.01 M, and 0.02 M).

### Synthesis of F-CoO NNAs
Typically, F-CoO NNAs were prepared by annealing Co(OH)F NNAs at 300 °C for 4 h with a heating rate of 2 °C min⁻¹.

### Synthesis of Co(OH)₂ NNAs
In a typical synthesis, a mixture of 0.125 M Co(NO₃)₂·6H₂O and 0.625 M urea was transferred into a 25 mL Teflon-lined stainless steel autoclave containing pretreated NF. The stainless steel autoclave was then maintained at 120 ℃ for 6 h to grow Co(OH)₂ NNAs on NF. Eventually, the product was rinsed with DI water and anhydrous ethanol, and dried in an oven at 60 °C for 12 h.

### Synthesis of Fe-CoO NNAs
Briefly, the synthetic procedure of Fe-CoO NNAs was similar to that of Fe, F-CoO NNAs except using Co(OH)₂ NNAs rather than Co(OH)F NNAs as the precursor.

### Synthesis of CoO NNAs
Briefly, the as-obtained Co(OH)₂ NNAs on NF were annealed at 300 °C for 4 h with a heating rate of 2 °C min⁻¹ to prepare CoO NNAs.

### Characterization
FESEM and EDS were conducted on A Hitachi S-4800 scanning electron micro-analyzer. TEM, HRTEM, and SAED images were acquired on a JEM-2100F field emission micro-analyzer at 200 kV. XRD was performed on a Philips PW3040/60 X-ray diffractometer equipped with Cu-Ka radiation. Ex situ Raman was conducted on a Renishaw in Via-Refles with a 532 nm laser as the exciting source. XPS was carried out on an ESCALab MKII X-ray photoelectron spectrometer using Al Ka X-ray radiation as the excitation source. In situ optical microscope pictures were captured by a BX53MRF-S optical microscope. O K-edge and Co K-edge XAS spectra were collected on the RapidXAFS 2 M (Anhui Absorption Spectroscopy Analysis Instrument Co., Ltd.), and the Si (533) spherically bent crystal analyzer with a radius of curvature of 500 mm was used for Co.

### Electrochemical measurements
All the electrochemical measurements were performed on a CHI 760E electrochemical workstation. The OER performances were evaluated in a standard three-electrode electrochemical system at room temperature with O₂-saturated 1 M KOH aqueous solution as the electrolyte. The as-prepared sample was used as the working electrode, while a carbon rod and a saturated calomel electrode served as the counter and reference electrodes, respectively. Before the assessment of OER performance, all the catalysts were activated to a stable state with cyclic voltammetry (CV) scanning at a scanning rate of 50 mV s⁻¹. LSV was performed at a sweep rate of 5 mV s⁻¹ and all the LSV curves were corrected with 90% IR compensation to eliminate the effect of solution resistance. Tafel plots were obtained from the polarization curves (Potential vs. the logarithm of current density (log |j|)). The electrochemical stability of the catalysts was measured by chronopotentiometry under the industrial-level current densities of 100 mA cm⁻² and 500 mA cm⁻². The potential versus RHE was converted according to the following equation.

$$E_{RHE} = E_{SCE} + 0.241 + 0.059pH \qquad (1)$$

The ECSAs of the catalysts were estimated based on the electric double-layer capacitance ($C_{dl}$). Cyclic voltammetry was performed from 0.15 to 0.25 V vs. RHE at various scanning rates from 20 to 100 mV s⁻¹. The $C_{dl}$ is estimated by the quotient of the current density difference ($\Delta j/2$) in the center of the potential window and the scanning rate. The calculation for ECSA is shown as follows.

$$ESCA = C_{dl}/C_S \times A \qquad (2)$$

where $C_s$ is 0.04 mF cm⁻² in alkaline electrolyte, and $A$ is the area of the working electrode (0.5 cm⁻² in this work). EIS measurements were performed at frequencies ranging from 100000 to 0.01 Hz under an AC voltage of 5 mV.

Electrochemical overall water splitting tests were measured using a two-electrode system. The as-prepared catalysts and Pt/C on NF were used as an anode for OER and a cathode for HER, respectively. For preparing the cathode, 5 mg of commercial 20 wt.% Pt/C catalyst and 40 µL of 5 wt.% Nafion solution were dispersed in 960 µL alcohol and water (1:1) by sonication for 30 min to form a homogeneous ink. Then the as-prepared Pt/C ink (475 µL) was dropped on the surface of NF. LSV curves were performed in 1 M KOH solution at a sweep speed of 5 mV s⁻¹ with 90% IR compensation. Electrochemical stability was measured by the chronopotentiometry method at the current density of 100 ~ 500 mA cm⁻².

Theoretical simulations, finite element analysis numerical simulations, and in situ electrochemical Raman measurements, as well as supplementary figures and tables, were provided in Supplementary Information.

## Data availability
The authors declare that the data supporting the findings of this study can be found in the paper and Supplementary Information files, or are available from the corresponding authors upon request.

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

## Acknowledgements

We thank the Anhui Absorption Spectroscopy Analysis Instrument Co, Ltd. for XAFS measurements and analysis. This work was financially supported by the National Natural Science Foundation of China (22272150), the Major Program of Zhejiang Provincial Natural Science Foundation of China (LD22B030002) and Zhejiang Provincial Ten Thousand Talent Program (2021R51009). H. W. acknowledges National Natural Science Foundation of China (22302177) and Key Science and Technology Project of Jinhua City (2020-1-044). H. H. acknowledges National Natural Science Foundation of China (22002083) and MSCA-IF-2020 Individual Fellowships (101024758).

## Author contributions

Y.H. and H.Y.W. conceived the original concept, supervised this project, and wrote the paper. P.C.Y. and K.Q.F. performed the synthesis, characterizations, and electrochemical performances of the catalysts. P.C.Y. and K.Q.F. analyzed the data. Y.H.W. helped the in situ Raman measurements. H.H. performed the density functional theory calculations. C.B.M. helped the preparation of catalysts. J.Q.N. helped with the data analysis and revision of the manuscript. All authors discussed the results and commented on the manuscript.

## Competing interests

The authors declare no competing interests.
