## [Peer Review File · Nature Communications]

REVIEWER COMMENTS

Reviewer #1 (Remarks to the Author):

Comments to the Author:

In this manuscript, the authors synthesized and tested an Fe and F co-doped CoO nanoneedle array electrode with rough surface, demonstrating the importance of the multi-scale regulation strategy to promote oxygen evolution reaction (OER) by adjusting the lattice oxygen chemistry and local electric field via the method of dual doping of cations and anions. However, I am afraid this work is not appropriate for the publication in Nature Communications due to the lack of novelty. The two mechanisms here have both been reported before and the combination of the two does not show any unexpected results or new findings. The performance of the electrode is not a state-of-the-art one. The specific comments are as follows:

1. In the abstract, we note that this concept "a synergy of tip enhancement and proximity effect" has been mentioned many times in previous works and lacks novelty in the field of OER. The authors should stress on the real advancement.
2. The authors need to explain why the addition of F activates lattice oxygen and thus awakens the lattice oxygen oxidation mechanism (LOM) pathway, rather than the addition of other nonmetallic elements such as N that have similar atomic dimensions to Co and O.
3. The authors stated that the nanoneedle structure of Co(OH)F is maintained after treatment with K₃[Fe(CN)₆]. As can be seen from Supplementary Fig. 2(a), the tip of the original Co(OH)F nanostructure has disappeared, showing a nanorod structure instead of a nanoneedle structure.
4. The figure quality in this manuscript is not good, which is not conducive to reading, please pay attention to this detail, which will be beneficial to the future submission.
5. The authors need to explain why the morphology changed from smooth to rough after the addition of Fe, but the morphology changed while the phase remained unchanged, which is rare in previous reports. (Page 7, line 135)
6. There are spelling and typo problems in the manuscript. (for example, Page 8, line 172)
7. The authors need to provide a more complete electrochemical testing process, such as testing the i_r compensation value of the LSV curve.
8. The authors claimed that "the introduction of Fe and F causes more unoccupied metal-oxygen bands to enter above E_f (shaded portion)", however this information cannot be obtained from the figure, and a larger picture of the shaded areas is needed. (Page 14, line 296)
9. Supplementary Fig. 5 shows that nanoneedles after F-doping seem to become larger in size. Is this correct?
10. The authors need to conduct longer stability tests to prove that Fe and F-CoO NNAs have outstanding long-term stability.

Reviewer #2 (Remarks to the Author):

Ye et al. demonstrated a conceptual strategy of coupling lattice oxygen and local electric field in the field of OER electrocatalyst, which is very interesting. And the overpotential at 10 mA cm⁻² is 169 mV, which is one of the best so far. However, the experiment evidence for lattice oxygen and local electric field is insufficient. And there were some issues in their simulation, which is suggested to reconduct. Hence, a major revision is suggested.

1, the characterization for lattice oxygen

a), Non-bonding oxygen states and direct O-O coupling is the evidence for LOM. In this work, the signal of O-O coupling in Raman data is too weak, which could not verify O-O coupling is formed. Here, ¹⁸O isotope technology or operando O XAFS is highly suggested.

b), LOM is a type of oxygen redox activity and during OER process, there should be no variations of Co valence. Why there is Co valence increase in their Raman data.

2, The origin for LOM mechanism is unclear

In this work, the authors claimed that doped Fe and F cooperatively tailor the electronic structure of CoO, leading to LOM. However, there was no experiment results for this. The readers might want to see how the dopant changes the local environment of Co atoms, which could be verified using Co XAFS and O K XAFS. The results of XAFS should be consistent with simulation, especially for the bond length and electronic states.

3, the local electric field

a), The local electric field is usually studied in the quantum scale (less than 10 nm). However, their sample is larger than 50 nm. The local electric field might be not suitable for this work.

b), their simulation in Figure 5 is less than 10 nm. It is also suggested to reconduct the simulation.

4, Simulation

a), The authors should provide the pDOS of non-bonding oxygen states, which usually work as evidence for LOM.

b), In the LOM pathway, there was no O-O coupling. The authors should reconduct the simulation.

5, the resolution of the figure

The authors should provide high-resolution figure.

Reviewer #3 (Remarks to the Author):

Promotion of water oxidation remains a challenge that must be addressed for renewables to meet a major fraction of global energy demand. This study proposes a strategy which utilizes electric-field and Fe/F doping to promote water oxidation in the context of CoO electrocatalyst.

1. The authors did not elaborate much on some aspects of their computational modeling. For example, there is no mention of the deployed DFT code and the flavor of DFT electrochemistry-based technique. Is the latter based on the Norskov's computational hydrogen electrode technique?
2. It appears that an extended low index facet model is used here to model reactions at the surfaces of the CoO nanostructures. While the use of such flat model should not be taken for granted, it does not account for the high curvature structures that may be present on the active site. The authors themselves mention that it is the high-curvature feature that can effectively promote efficiency and improve current density in different electrocatalytic reactions (p. 4). In particular, Fe doping is mentioned to be highly influential in the generation of such morphology (p. 7). If CoO possesses a fcc-bulk structure, why not consider using the (211) slab to model a high curvature active site.
3. The models for the doped CoO contain a single Fe and/or F atom situated on the top site. Is there any evidence that points to preferential dopant incorporation in the near surface region. While XRD pattern supports the incorporation of Fe and F in the CoO lattice, it does not approximately pinpoint where the dopants are in the nanoneedles. In the absence of experimental confirmation, maybe DFT thermodynamics analysis can help sort this out (see Phys. Chem. Chem. Phys, 21, 23626-23637 (2019)).
4. A particular highlight of this work is high curvature of the Cu nanoneedles results in a large local electric field on their tips which can promote the reaction. In this work, finite element simulation method is used to explore this further. Another way to look at the role of the electric field is to use DFT. The authors must have used VASP in this work, a software that allows reaction modeling in the presence of electric field.

We thank the reviewers for their time and very useful comments in improving the quality of this manuscript. Provided below is our detailed response to each question.

Reviewers' comments:

Reviewer #1: In this manuscript, the authors synthesized and tested an Fe and F co-doped CoO nanoneedle array electrode with rough surface, demonstrating the importance of the multi-scale regulation strategy to promote oxygen evolution reaction (OER) by adjusting the lattice oxygen chemistry and local electric field via the method of dual doping of cations and anions. However, I am afraid this work is not appropriate for the publication in Nature Communications due to the lack of novelty. The two mechanisms here have both been reported before and the combination of the two does not show any unexpected results or new findings. The performance of the electrode is not a state-of-the-art one.

Response:

Thank you for your feedback and raising the concern regarding the novelty of our study. We would like to address your comment and explain why our study still contributes to the existing knowledge and advances the field. While the LOM and tip-enhanced electric field have been investigated in the previous reports, there are several important distinctions and contributions to the novelty of our study.

Distinctions and novelty: The reactive rate of OER is highly related to the electrochemical reaction process and mass transfer to the electrode surface. Thus, regulating either aspect alone has a certain limitation. For instance, the reported OER catalysts via LOM can exhibit high activity at low current density (usually 10 mA cm^{-2}), but inferior performance at industrial current density ($>200 \text{ mA cm}^{-2}$). Developing an efficient strategy that can regulate lattice oxygen redox chemistry for accommodating LOM and construct abundant high-curvature sites to enhance local electric field simultaneously is of significance for OER but extremely challenging. Hitherto, there is still no report on this field.

Importantly, previous works on LOM catalysts mainly focused on perovskite, multimetallic alloy, and metallic (oxy)hydroxide. Cobalt oxides have been regarded as promising candidates for OER. However, how to switch the OER mechanism of cobalt oxides from conventional AEM to LOM has been rarely studied. In addition, the cation and anion dual doping strategy for triggering LOM by tailoring metal cations and

oxygen ligands simultaneously has never been reported as far as we know. The working mechanism of cation and anion dopants has yet to be elucidated. Moreover, the tip-enhanced electric field has rarely been studied in OER and the enhanced mechanism was mainly attributed to electrolyte enrichment. The synergy of tip enhancement and proximity effect on OER, especially their effect on the OER process such as intermediate formation and O₂ gas release has yet to be clarified.

Findings: Combining X-ray absorption spectroscopy, *in situ* Raman spectroscopy, chemical probe, ¹⁸O isotope technology, differential electrochemical mass spectrometry, density functional theory calculations, finite element method simulations, and *in situ* optical microscope, the disparate and synergistic effects of Fe and F dopants on the electronic structure and OER mechanism, as well as the effect on the morphology, local electric field, OH⁻ ion distribution, reaction intermediates and O₂ release were intensively explored. The results show that Fe and F dopants mainly adjust the electronic states of the Co atom and oxygen ligand, respectively, leading to the disparate electrochemical behaviors and OER mechanisms in Fe-CoO NNAs and F-CoO NNAs. Compared to Fe-CoO NNAs, F-CoO NNAs, and CoO NNAs, the Fe, F-CoO NNAs have stronger metal-oxygen covalency and higher lattice oxygen reactivity, thus facilitating the electron transfer and awakening the LOM pathway. Meanwhile, it is confirmed that the rough needle-shaped structure induced by Fe doping possesses a sharp tip enhancement and a proximity effect, contributing to a stronger electric field to favor the enrichment of OH⁻ around the active sites and promote the release of O₂ bubbles. More importantly, it has been found that the enhanced local electric field in Fe, F-CoO NNAs can optimize the energy barrier of LOM pathway. Correspondingly, the thermodynamics and kinetics for OER are promoted simultaneously, leading to boosted electrocatalytic performance.

Electrocatalytic Performance: the as-prepared Fe, F-CoO NNAs can deliver a current density of 10 mA cm⁻² at the low overpotential of 169 mV. Besides, only overpotentials of 234 mV and 277 mV are required to deliver large current densities of 100 and 500 mA cm⁻², respectively. In addition, it can also be stable for 300 h at large current densities of 100 mA cm⁻² and 500 mA cm⁻², suggesting its long-term OER stability. These results are among the best of the state-of-the-art catalysts (Supplementary Tables 3 and 5).

In summary, this work provides new insights into heteroatom doping-induced lattice oxygen redox chemistry and local electric field enhancement, which open a new

path to design high-efficiency catalysts for enhancing OER, and also deepens the understanding of the structure-activity relationship and electrocatalytic mechanism. This is an important and central contribution of this study. We believe this work could offer some promising views for other catalytic reactions as well as energy storage systems that involve oxygen chemistry or mass transfer kinetics.

We realize that the novelty may not have been expressed in original manuscript, so we have made improvements in the introduction to clearly show the limitations of current research and highlight the distinctions and advances in this work (see Pages 4 and 5).

Supplementary Table 3. Comparison of the overpotentials of Fe, F-CoO NNAs with the state-of-the-art transition metal-based OER electrocatalysts at various current densities (10 mA cm⁻², 100 mA cm⁻², and 500 mA cm⁻²).

Catalysts	η_{10} (mV)	η_{100} (mV)	η_{500} (mV)	References
Fe, F-CoO NNAs	169	234	277	This work
LSC/K-MoSe ₂	230	/	/	Nat. Commun. 2021, 12, 4606.
S-FeOOH/IF	244	308	/	Adv. Funct. Mater. 2022, 2112674
Fe-doped-(Ni-MOFs)/FeOOH	/	278	/	Angew. Chem. Int. Ed. 2022, 61, e202116934.
CF-FeSO	192	230	/	Nat. Commun. 2022, 13, 605.
CoSe ₂ -D _{Fe} -V _{Co}	/	520	/	Nat. Commun. 2020, 11, 1664.
W-NiS _{0.5} Se _{0.5}	171	239	/	Adv. Mater. 2022, 34, 2107053.
CoFe-LDH/GF	252	285	/	Chin. Chem. Lett. 2022, 33, 890-892.
V-CoP ₂ /CC	91	498	/	Angew. Chem. Int. Ed. 2022, 61, e202116233.
Fe _{0.4} Ni _{0.6} -alloy fiber paper	/	287	/	Appl. Catal. B: Environ. 2021, 286, 119902.
CF/VMFO	/	/	225	Nat. Commun. 2021, 12, 1380.
FeNi(VO ₄) _x @NF	/	274	/	Small 2020, 16, 2002412
NiMoFeO@NC	/	290	/	Matter 2020, 3, 2124-2137.
Fe _{0.052} Ni-POMo	255.3	295	/	Adv. Funct. Mater. 2021, 31, 2101792
NiFe-Boride	167	/	/	Nat. Commun. 2021, 12, 6089.
Ni _x Fe _y Co _{6-x-y} Mo ₆ C/NF	212	275	336	Appl. Catal. B: Environ. 2021, 290, 120049.
NiFeB	/	252	/	Nat. Commun. 2022, 13, 6094.

NiMo-Fe	217	264	/	Appl. Catal. B: Environ. 2022, 307, 121150.
NFO-S5	/	232	/	Appl. Catal. B: Environ. 2022, 305, 121030
(Ni,Co)Se ₂ -GA	250	340	/	ACS Catal. 2017, 7, 6394.
Fe-CoP/NF	190	/	295	Adv. Sci. 2018, 5, 1800949
EBP/CoFeB	227	313	/	ACS Nano 2021, 15, 12418–12428

Supplementary Table 5. Comparison of the electrocatalytic stability of Fe, F-CoO NNAs with the reported transition metal-based OER electrocatalysts at large current densities.

Catalysts	Stability	Current density (mA cm ⁻²)	References
Fe, F-CoO	300 h	100	This work
	300 h	500	
NiFeB	130 h	500	Nat. Commun. 2022, 13, 6094.
CF/VMFO	30 h	250	Nat. Commun. 2021, 12, 1380.
NiMoN/NiFe LDH	250 h	1000	Nat. Commun. 2023, 14, 1873.
MIL-53(Fe)-2OH	100 h	100	Adv. Mater. 2023, 35, 2208904.
CoFePS	42 h	500	Adv. Funct. Mater. 2023, 2308422.
FeCoSn(OH) ₆ -300	200 h	100	Nat. Commun. 2022, 13, 1187.
SO-NFO NS _{D21}	24 h	100	ACS Energy Lett. 2023, 8, 3504.
Co(OH)(CO ₃) _{0.5}	30 h	100	ACS Catal. 2023, 13, 8821.
Zn, S-Fe ₃ O ₄ -FeOOH/IF	100 h	500	Adv. Funct. Mater. 2023, 2303776.
Ni-BDC-1R	100 h	100	Angew. Chem. Int. Ed. 2022, 134, e202214794.

(1) In the abstract, we note that this concept “a synergy of tip enhancement and proximity effect” has been mentioned many times in previous works and lacks novelty in the field of OER. The authors should stress on the real advancement.

Response:

We are grateful to the reviewer for the valuable suggestion. The real advances of this work are as follows:

While the synergy of tip enhancement and proximity effect has been studied in electrocatalytic reactions such as CO₂ electroreduction, such a synergy concept has not been studied in OER as far as we know. Moreover, previous studies on the mechanism of tip electric field in OER is focused on electrolyte enrichment. However, the effect on the OER process such as the formation of oxygen-containing intermediates and the release of O₂ gas has yet to be clarified before.

More importantly, considering the reaction rate is highly related to the electrode reaction and mass transfer to the electrode surface, we herein propose a facile and effective cation and anion dual doping strategy to regulate the lattice oxygen redox chemistry and enhance the local electric field simultaneously, and also demonstrate their important cooperative role in promoting OER. To our knowledge, this strategy and concept has never been reported in this field. In addition, to get insight into the role of Fe and F dual doping in the electrochemical behavior of CoO, the disparate and synergistic effects of Fe and F dopants on the electronic structure and OER mechanism, as well as the effect on the morphology, local electric field, OH⁻ ion distribution, reaction energy barrier, and O₂ release were intensively explored by X-ray absorption spectroscopy, *in situ* Raman spectroscopy, chemical probe, oxygen isotope technology, differential electrochemical mass spectrometry, density functional theory calculations, finite element method simulations, and *in situ* optical microscope. The results show that Fe and F dopants mainly adjust the electronic states of the Co atom and oxygen ligand, respectively, resulting in disparate electrochemical behaviors and OER mechanisms in Fe-CoO NNAs and F-CoO NNAs. Fe and F dual dopants jointly upshift Co d-band center (ϵ_d) and O 2p band around the Fermi level (E_f), improve the covalency of the metal-oxygen band as well as activate lattice oxygen, thus facilitating the electron transfer and awakening the LOM pathway. In addition, it is confirmed that the rough

needle-shaped structure induced by Fe doping possesses a sharp tip enhancement and a proximity effect, contributing to a stronger electric field to favor the enrichment of OH⁻ around the active sites as well as promote the release of O₂ bubbles. More importantly, it is found that the enhanced local electric field in Fe, F-CoO NNAs can optimize the energy barrier of LOM pathway, thus improving the OER activity.

Last but not the least, rough arrays of Fe, F-CoO NNAs catalyst exhibit excellent OER performance, including low overpotentials of 169 mV at 10 mA cm⁻² and 277 mV at 500 mA cm⁻² as well as long-term stability (300 h at 100 mA cm⁻² and 500 mA cm⁻²). These results are among the best of the state-of-the-art OER catalysts, demonstrating the great potential of Fe, F-CoO NNAs in industrial electrocatalytic water oxidation.

In summary, this work provides new insights into heteroatom doping-induced lattice oxygen redox chemistry and local electric field enhancement, which may open a new path to design high-efficiency catalysts for accelerating the OER kinetics, and also deepens the understanding of the structure-activity relationship and electrocatalytic mechanism. More broadly, this work could offer some promising views for other catalytic reactions as well as energy storage systems that involve oxygen chemistry or mass transfer kinetics. We have made improvements in the introduction to clearly show the limitations of current research in this area and highlight the advances of our work (see Pages 4 and 5).

(2) The authors need to explain why the addition of F activates lattice oxygen and thus awakens the lattice oxygen oxidation mechanism (LOM) pathway, rather than the addition of other nonmetallic elements such as N that have similar atomic dimensions to Co and O.

Response:

We are grateful to the reviewer for the thoughtful suggestion. Given the large electronegativity of oxygen ($x = 3.44$), typical solid-state materials exhibit a large charge transfer energy with strong ionic character of the metal–oxygen bond. Thus, the oxygen ligands can be constrained in the lattice matrix. To allow the lattice oxygen activation and awaken the LOM pathway, it is necessary to modulate the absolute energy level of the O 2p band and also tune the metal–O bond covalency. Although F and other nonmetallic elements such as N have similar size compatibility to O, their electronegativities are different, which have a different influence on the electronic state of O once doped into the lattice. It is noteworthy that the electronegativity of N ($x =$

3.04) is lower compared to that of O ($x = 3.44$), whereas the electronegativity of F ($x = 3.98$) is larger and electrons are more easily attracted by F than O. Therefore, F doping can decrease the valence electron density of O^{2-} and may lead to more active lattice oxygen ions (see related Refs. *J. Mater. Chem. A*, 2020, 8, 14091; *Appl. Catal. B: Environ.* 2022, 309, 121236). The theoretical calculations further prove this assumption, which demonstrate that F doping in CoO can improve the O 2p band around the Fermi level and enhance the metal-oxygen covalency, as well as favor the formation of oxygen vacancy in the LOM pathway, thus awakening the LOM pathway. The related discussion has been added in the revised manuscript (see Page 5).

(3) The authors stated that the nanoneedle structure of Co(OH)F is maintained after treatment with $K_3[Fe(CN)_6]$. As can be seen from Supplementary Fig. 2(a), the tip of the original Co(OH)F nanostructure has disappeared, showing a nanorod structure instead of a nanoneedle structure.

Response:

We are grateful to the reviewer for pointing this out. The large-scale SEM image (Supplementary Fig. 2a) shows that the products after treatment with $K_3[Fe(CN)_6]$ mostly maintain the nanoneedle structure of Co(OH)F. Some tips were broken probably due to ultrasound during the preparation of the samples. We have modified the description in the revised manuscript (see Page 6).

Supplementary Fig. 2. (a) FESEM image of the $Fe[(CN)_6]^{3-}$ modified Co(OH)F NNAs.

(4) The figure quality in this manuscript is not good, which is not conducive to reading, please pay attention to this detail, which will be beneficial to the future submission.

Response:

We are grateful to the reviewer for pointing this out. High-resolution figures have been provided in the revised manuscript and Supplementary Information.

(5) The authors need to explain why the morphology changed from smooth to rough after the addition of Fe, but the morphology changed while the phase remained unchanged, which is rare in previous reports. (Page 7, line 135)

Response:

We are grateful to the reviewer for the valuable suggestion. After annealing treatment, arrays of Fe and F co-doped CoO nanoneedles are constructed, which have a rather rough surface. This may be because the adsorbed iron ions on the external surfaces *in situ* substitute some cobalt ions, leading to the formation of more structural defects in Fe, F-CoO NNAs (see related Refs. Nat. Commun. 2022, 13, 2591; Catalysts 2019, 9, 458). This is further verified by the enhanced surface roughness of Fe, F-CoO NNAs with the increase of Fe dopant (Supplementary Fig. 4). The related discussion has been added in the revised manuscript (see Page 6).

As it can be seen, the surfaces of the nanoneedle arrays become rough after Fe is incorporated into CoO, while no impurities except for cubic phase CoO are formed. This is probably due to the similar ion radius of Fe³⁺ with Co²⁺ and the low concentration of Fe³⁺ in the lattice (0.6 at.%) (see related Refs. Appl. Surf. Sci. 2018, 454, 46; Catalysts 2019, 9, 458; Chem. Eng. J., 2021, 424, 130400). This phenomenon that the morphology is changed while the phase is maintained is very common for doped nanomaterials (see related Refs. Nat. Commun. 2022, 13, 2591; Sci China Mater., 2021, 64, 1889; Catalysts 2019, 9, 458; Appl. Surf. Sci. 2018, 454, 46).

(6) There are spelling and typo problems in the manuscript. (for example, Page 8, line 172)

Response:

We are grateful to the reviewer for pointing this out. We have carefully revised the spelling and typo errors throughout the manuscript.

(7) The authors need to provide a more complete electrochemical testing process, such as testing the iR compensation value of the LSV curve.

Response:

We are grateful to the reviewer for the valuable suggestion. According to your suggestion, we have added more details for electrochemical testing (see Pages 21 and 22).

Electrochemical measurements. All the electrochemical measurements were

performed on a CHI 760E electrochemical workstation. The OER performances were evaluated in a standard three-electrode electrochemical system at room temperature with O₂-saturated 1 M KOH aqueous solution as the electrolyte. The as-prepared sample was used as the working electrode, while a carbon rod and a saturated calomel electrode served as the counter and reference electrodes, respectively. Before the assessment of OER performance, all the catalysts were activated to a stable state with cyclic voltammetry (CV) scanning at a scanning rate of 50 mV s⁻¹. LSV was performed at a sweep rate of 5 mV s⁻¹ and all the LSV curves were corrected with 90% IR compensation to eliminate the effect of solution resistance. Tafel plots were obtained from the polarization curves (Potential vs. the logarithm of current density (log |j|)). The electrochemical stability of the catalysts was measured by chronopotentiometry under the industrial-level current densities of 100 mA cm⁻² and 500 mA cm⁻². The potential versus reversible hydrogen electrode (RHE) was converted according to the following equation.

$$E_{\text{RHE}} = E_{\text{SCE}} + 0.241 + 0.059\text{pH} \quad (1)$$

The ECSAs of the catalysts were estimated based on the electric double-layer capacitance (C_{dl}). Cyclic voltammetry was performed from 0.15 to 0.25 V vs. RHE at various scanning rates from 20 to 100 mV s⁻¹. The C_{dl} is estimated by the quotient of the current density difference (Δj/2) in the center of the potential window and the scanning rate. The calculation for ECSA is shown as follows.

$$\text{ECSA} = C_{\text{dl}} / C_{\text{s}} \times A \quad (2)$$

where C_s is 0.04 mF cm⁻² in alkaline electrolyte, and A is the area of the working electrode (0.5 cm² in this work). Electrochemical impedance spectroscopy (EIS) measurements were performed at frequencies ranging from 100000 to 0.01 Hz under an AC voltage of 5 mV.

Electrochemical overall water splitting tests were measured using a two-electrode system. The as-prepared catalysts and Pt/C on NF were used as an anode for OER and a cathode for HER, respectively. For preparing the cathode, 5 mg of commercial 20 wt.% Pt/C catalyst and 40 μL of 5 wt.% Nafion solution were dispersed in 960 μL alcohol and water (1:1) by sonication for 30 min to form a homogeneous ink. Then the as-prepared Pt/C ink (475 μL) was dropped on the surface of NF. LSV curves were performed in 1 M KOH solution at a sweep speed of 5 mV s⁻¹ with 90% IR compensation. Electrochemical stability was measured by the chronopotentiometry

method at the current density of $100 \sim 500 \text{ mA cm}^{-2}$.

(8) The authors claimed that "the introduction of Fe and F causes more unoccupied metal-oxygen bands to enter above E_f (shaded portion)", however this information cannot be obtained from the figure, and a larger picture of the shaded areas is needed. (Page 14, line 296)

Response:

We are grateful to the reviewer for the valuable suggestion. According to your suggestion, a magnified image of the shaded areas has been provided in the insets of Figs. 5c and 5d.

Figure 5 DFT calculations. (c) Co 3d, O 2p, F 2p, and Fe 3d PDOS spectra of the Fe, F-CoO slab with an inset of the shaded area. (d) Co 3d and O 2p PDOS spectra of the CoO slab with an inset of the shaded area.

(9) Supplementary Fig. 5 shows that nanoneedles after F-doping seem to become larger in size. Is this correct?

Response:

We are grateful to the reviewer for the valuable suggestion. The sizes of CoO NNAs and F-CoO NNAs were calculated based on the SEM images. As shown in Fig. R1, the CoO NNAs and F-CoO NNAs show similar sizes with the base diameter centered around 90~120 nm.

Figure R1 FESEM images of the (a) CoO NNAs and (c) F-CoO NNAs. The size distribution of the (b) CoO NNAs and (d) F-CoO NNAs.

(10) The authors need to conduct longer stability tests to prove that Fe and F-CoO NNAs have outstanding long-term stability.

Response:

We are grateful to the reviewer for the valuable suggestion. According to your suggestion, we have conducted longer stability tests of Fe, F-CoO NNAs. As shown in **Fig. 3e**, the real-time potential of Fe, F-CoO NNAs hardly increases during the continuous testing for 300 h at large current densities of 100 mA cm^{-2} and 500 mA cm^{-2} , suggesting their excellent electrocatalytic stability and great potential in practical applications. This value is also comparable or even superior to most of the transition metal-based OER electrocatalysts, highlighting the outstanding long-term stability of Fe, F-CoO NNAs catalyst (**Supplementary Table 5**). The related discussion has been added to the revised manuscript (**see Page 10**).

Figure 3 (e) Chronopotentiometry measurements of the Fe, F-CoO NNAs at large current densities of 100 and 500 mA cm⁻².

Supplementary Table 5. Comparison of the electrocatalytic stability of Fe, F-CoO NNAs with the reported transition metal-based OER electrocatalysts at large current densities.

Catalysts	Stability	Current density (mA cm ⁻²)	References
Fe, F-CoO	300 h	100	This work
	300 h	500	
NiFeB	130 h	500	Nat. Commun. 2022, 13, 6094.
CF/VMFO	30 h	250	Nat. Commun. 2021, 12, 1380.
NiMoN/NiFe LDH	250 h	1000	Nat. Commun. 2023, 14, 1873.
MIL-53(Fe)-2OH	100 h	100	Adv. Mater. 2023, 35, 2208904.
CoFePS	42 h	500	Adv. Funct. Mater. 2023, 2308422.
FeCoSn(OH) ₆ -300	200 h	100	Nat. Commun. 2022, 13, 1187.
SO-NFO NS _{D21}	24 h	100	ACS Energy Lett. 2023, 8, 3504.
Co(OH)(CO ₃) _{0.5}	30 h	100	ACS Catal. 2023, 13, 8821.

Zn, S-Fe ₃ O ₄ -FeOOH/IF	100 h	500	Adv. Funct. Mater. 2023, 2303776.
Ni-BDC-1R	100 h	100	Angew. Chem. Int. Ed. 2022, 134, e202214794.
(Act)-(Ni,Mn)- (Co) _{tet} (Co ₂) _{oct} O ₄ NSs	100 h	100	Angew. Chem. Int. Ed. 2023, 62, e202214600.

Reviewer #2: Ye et al. demonstrated a conceptual strategy of coupling lattice oxygen and local electric field in the field of OER electrocatalyst, which is very interesting. And the overpotential at 10 mA cm⁻² is 169 mV, which is one of the best so far. However, the experiment evidence for lattice oxygen and local electric field is insufficient. And there were some issues in their simulation, which is suggested to reconduct. Hence, a major revision is suggested.

(1) the characterization for lattice oxygen. a), Non-bonding oxygen states and direct O-O coupling is the evidence for LOM. In this work, the signal of O-O coupling in Raman data is too weak, which could not verify O-O coupling is formed. Here, ¹⁸O isotope technology or operando O XFAS is highly suggested. b), LOM is a type of oxygen redox activity and during the OER process, there should be no variations of Co valence. Why there is Co valence increase in their Raman data.

Response:

We are grateful to the reviewer for the valuable suggestion. a) As suggested, ¹⁸O isotope technology has been carried out. According to the previous reports (see related Refs. Angew. Chem. 2019, 131, 1040; Energy Environ. Sci. 2021, 14, 4647), due to the isotope effect, when ¹⁶O is substituted with a larger mass atom ¹⁸O, it will cause a red-shifted phenomenon for the vibrational bond peak in Raman spectroscopy. Therefore, similar to the aforementioned works (see related Refs. Angew. Chem. 2019, 131, 1040; J. Am. Chem. Soc. 2020, 142, 11901), the oxygen isotope technology was first coupled with electrochemical *in situ* Raman spectroscopy to identify the LOM reaction mechanism. A three-step strategy is adopted for the oxygen isotope labelling experiments (Fig. 4f). First, the as-prepared Fe, F-CoO NNAs catalyst was labelled in 0.1 M KOH H₂¹⁸O aqueous solution (denoted as ¹⁸O-KOH) using a chronoamperometry method for 20 min at 1.664 V versus RHE and then rinsed by water. The ¹⁸O-KOH electrolyte was further replaced with 0.1 M KOH H₂¹⁶O (¹⁶O-KOH) aqueous solution.

At last, *in situ* Raman was carried out on the ^{18}O -labelled Fe, F-Co ^{18}O NNAs from 1.23 to 1.43 V versus RHE in ^{16}O -KOH electrolytes. As indicated in Supplementary Fig. 24, after ^{18}O labelling of Fe, F-CoO NNAs, the F_{2g} and A_{1g} peaks of the Co–O band both shift to lower frequencies, indicating that the successful labelling of lattice oxygen with ^{18}O . Furthermore, *in situ* Raman spectra of Fe, F-Co ^{18}O NNAs in ^{16}O -KOH electrolytes reveal that the peaks of F_{2g} mode of Co–O band and OO^- band gradually moved toward higher frequencies with the increase of potential from 1.23 to 1.43 V, demonstrating that the lattice oxygen of Fe, F-CoO NNAs can exchange with the electrolyte and get involved in the OER process (Fig. 4g). Additionally, ^{18}O isotope technology combined with on-line differential electrochemical mass spectrometry (DEMS) was used to detect O_2 gas generated during the OER (Fig. 4f). As shown in Fig. 4h, the signals for mass-to-charge ratio (m/z) of 32 and 34, corresponding to $^{32}\text{O}_2$ ($^{16}\text{O}^{16}\text{O}$) and $^{34}\text{O}_2$ ($^{16}\text{O}^{18}\text{O}$), respectively, are detected for Fe, F-Co ^{18}O NNAs, confirming the involvement of lattice oxygen in the OER.

The above observations provide more compelling evidence for the LOM reaction mechanism. The related experiment details and results have been added in the revised supplementary information (see Pages S5 and S6) and the revised manuscript (see Pages 13).

Figure 4 (f) Schematic illustration of the oxygen isotope labelling experiments. (g) *In situ* Raman spectra and (h) DEMS spectra coupled with isotope experiment of ^{18}O labelled Fe, F-Co ^{18}O NNAs.

b) According to the previous reports (see related Refs. J. Energy Chem., 2022, 70, 373; Energy Environ. Sci., 2022, 15, 206), the OH^- electrolyte could oxidize the low-valence transition metal site (such as CoO) even if there were no electron transfers between cathode and anode. Therefore, the transformation of low-valence Co^{2+} cations to Co^{3+} is spontaneous and can occur rapidly even at open circuit potential. And based on the aforementioned works, when the potential is elevated, more Co^{2+} ions could be transformed into Co^{3+} , resulting in a higher oxidation state. The high valence state is favorable for lattice oxygen activation (see related Ref. Adv. Mater. 2022, 34, 2202523).

This dynamic variation from low-valence metal to high-valence metal can also be found in other transition metal-based catalysts with the LOM pathway when it involves a single-metal-site or dual-metal-site mechanism (see related Refs. *Adv. Mater.* 2022, 34, 2202523; *Nat. Commun.*, 2020, 11, 4066, *Energy Environ. Sci.*, 2021, 14, 4647). The related descriptions have been added to the revised manuscript (see Page 13).

(2) The origin for LOM mechanism is unclear. In this work, the authors claimed that doped Fe and F cooperatively tailor the electronic structure of CoO, leading to LOM. However, there was no experiment results for this. The readers might want to see how the dopant changes the local environment of Co atoms, which could be verified using Co XAFS and O K XAFS. The results of XAFS should be consistent with simulation, especially for the bond length and electronic states.

Response:

We are grateful to the reviewer for the valuable suggestion. According to your suggestion, XAS was conducted to explore the electronic and chemical coordination structures of the catalysts. Fig. 2a presents the O K-edge spectra of the CoO NNAs, F-CoO NNAs and Fe, F-CoO NNAs, in which the O K-edge pre-edge peak centered at 532 eV is attributed to the hybridization of O 2p-state with the Co 3d. As described, the O K-edge pre-peak almost disappears for F-CoO NNAs, suggesting that oxygen release occurs with F doping (see related Ref. *Nature* 2022, 666, 305). It is notable that compared with CoO NNAs, Fe, F-CoO NNAs present a more intense pre-edge peak, suggesting the strengthened metal-oxygen covalency (see related Refs. *Energy Environ. Mater.* 2021, 4, 246; *Adv. Funct. Mater.* 2021, 31, 2001633). This result is consistent with those from O 1s XPS spectra. The Co K-edge X-ray absorption near edge structure (XANES) spectra show that the absorption edges of the three samples are located between standard CoO and Co₂O₃, manifesting the valence state of Co between +2 and +3 (Fig. 2b). A shift to lower energies is observed for Fe, F-CoO NNAs compared to CoO NNAs, suggesting the lower Co valence. The Fourier-transformed extended X-ray absorption fine structure (FT-EXAFS) spectra of Co K-edge reveal two shells at ~1.4 and ~2.4 Å, corresponding to the Co–O and Co–Co scattering paths, respectively (Fig. 2c). The almost overlapped Co K-edge oscillation plots and the similar distances of Co–O and Co–Co in the undoped and doped samples indicate that the incorporation of Fe and F does not cause significant structural changes in CoO (Supplementary Figs. 10, 11 and Supplementary Table 2). This is further confirmed by the similar wavelet

transform (WT) plots (Figs. 2d-2i). The above results agree well with the observations from XRD, Raman and DFT simulations. The related descriptions have been added in the revised manuscript (see Pages 8 and 9).

Figure 2 Electronic and chemical coordination structures of the Fe, F-CoO NNAs and references. (a) Normalized O K-edge spectra of CoO NNAs, F-CoO NNAs, and Fe, F-CoO NNAs. (b) Normalized Co K-edge XANES spectra, (c) FT-EXAFS spectra of Co K-edge, and (d-i) WT-EXAFS analysis of CoO NNAs, F-CoO NNAs, Fe, F-CoO NNAs, and the standard references including Co foil, standard CoO and Co₂O₃.

Supplementary Fig. 10. Co K-edge EXAFS oscillation functions of CoO NNAs, F-CoO NNAs, Fe, F-CoO NNAs, and the standard references including Co foil, standard CoO and Co₂O₃.

Supplementary Fig. 11. Co K-edge EXAFS fits for the CoO NNAs, F-CoO NNAs, and Fe, F-CoO NNAs.

Supplementary Table 2. Structural parameters of CoO NNAs, F-CoO NNAs and Fe, F-CoO NNAs extracted from the EXAFS fitting. ($S_0^2=0.80$).

Scattering pair	CN	R (Å)	σ^2 (10^{-3}Å^2)	ΔE_0 (eV)	R factor	
CoO	Co-O	4.0±0.1	2.13±0.02	5.3±0.7	4.5±0.3	0.02
	Co-Co	5.1±0.9	3.01±0.02	5.6±0.9	-4.2±0.5	0.02
F-CoO	Co-O/F	3.9±0.5	2.13±0.02	6.4±0.7	4.5±0.5	0.02
	Co-Co	5.2±0.7	3.01±0.02	6.8±0.9	-4.2±0.5	0.02
Fe, F-CoO	Co-O/F	4.1±0.2	2.13±0.02	6.9±0.7	4.5±0.5	0.02
	Co-Co/Fe	5.1±0.5	3.01±0.02	7.7±0.3	-4.2±0.5	0.02

S_0^2 is the amplitude reduction factor $S_0^2=0.8$; CN is the coordination number; R is the interatomic distance (the bond length between central atoms and surrounding coordination atoms); σ^2 is Debye-Waller factor (a measure of thermal and static disorder in absorber-scatterer distances); ΔE_0 is edge-energy shift (the difference between the zero kinetic energy value of the sample and that of the theoretical model). R factor is used to value the goodness of the fitting.

(3) The local electric field. a), The local electric field is usually studied in the quantum scale (less than 10 nm). However, their sample is larger than 50 nm. The local electric field might be not suitable for this work. b), their simulation in Figure 5 is less than 10 nm. It is also suggested to reconduct the simulation.

Response:

We are grateful to the reviewer for the valuable suggestion. a) Previous studies have demonstrated that while the nanoneedle/nanocone samples are larger than 50 nm, their tips (usually less than 10 nm) can produce locally enhanced electrostatic field due to the high-curvature structure (see related Refs. Nature 2016, 537, 382; Sci. Adv. 2022, 8, eabm9477).

b) According to the reports (see related Refs. Nature 2016, 537, 382; Sci. Adv. 2022, 8, eabm9477), finite element analysis numerical simulation is used to explore the role of the rough nanoneedle structure and the effect on OER performance in our work. Four models with different radii and disparate surfaces were constructed to simulate the immersion of CoO electrodes in 1.0 M KOH electrolyte. The surface structure and the radii of the nanoneedles are constructed based on the SEM images. These models include a nanoneedle (top radius: 110 nm, bottom radius: 150 nm), a nanoneedle with a smooth surface (top radius: 3.2 nm, bottom radius: 60 nm), a nanoneedle (top radius: 3.2 nm, bottom radius: 60 nm) with 1.5 nm spheres on the surface. In addition, to explore the proximity effect, the 1.5 nm spheres with different distances (16 nm and 6 nm) were also constructed. The finite element simulation results are shown in Supplementary Fig. 33. To show the tip structure more clearly, the enlarged images of the surface positive charge density distribution and surface OH⁻ density distribution on the tips of the four electrodes are displayed in Fig. 6. Evidently, positive charge density is mainly centralized at the tips of nanoneedles and displays a 24-fold increase with the sharpening of the top radius from 110 nm to 3.2 nm (Figs. 6a and 6b). Intriguingly, a larger positive charge density of 0.094 C m⁻² is found, approximately 1.4 times that of the smooth one (Fig. 6c). Moreover, decreasing the distance between the spherules from 16 to 6 nm is favorable to converging the positive charge density surrounding the nanoneedle surface and increasing the charge density dramatically, suggesting a conspicuous proximity enhancement (Fig. 6d). The charge density can reach 0.11 C m⁻², corresponding an electric field of 1.58×10^{-2} V/Å. Besides, it is found that the OH⁻ concentration on rough needle-like structure with small spherule gaps (6 nm) reaches the highest, increased by 170 % and 20 % compared to smooth needle tips and the model with large spherule gaps (16 nm), respectively (Figs. 6e-6h). These results demonstrate that the rough nanoneedle-structured Fe, F-CoO arrays can induce the strongest local electric field and the largest OH⁻ ion concentration at the electrode surface as a result of the coupling of tip-enhanced electric field and the adjacency effect. These descriptions are shown in the revised supplementary information (see Page S3)

and the revised manuscript (see Pages 16 and 17).

Supplementary Fig. 33. Finite element simulations. (a-d) Surface positive charge density distribution and (e-h) surface OH^- density distribution on different electrode models. (a, e) The nanoneedles with a top radius of 110 nm and a bottom radius of 150 nm. (b, f) The nanoneedles with a top radius of 3.2 nm and a bottom radius of 60 nm. The nanoneedles (top radius: 3.2 nm and bottom radius: 60 nm) with 1.5 nm spherules on the surface, and the gaps between the spherules are (c, g) 16 nm and (d, h) 6 nm.

Figure 6 Enlarged images of (a-d) surface positive charge density distribution and (e-h) surface OH^- density distribution on the tips of different electrode models. (a, e) The nanoneedles with a top radius of 110 nm and a bottom radius of 150 nm. (b, f) The smooth nanoneedles with a top radius of 3.2 nm and a bottom radius of 60 nm. The nanoneedles (top radius: 3.2 nm and bottom radius: 60 nm) with 1.5 nm spherules on the surface, and the gaps between the spherules are (c, g) 16 nm and (d, h) 6 nm.

(4) Simulation. a), The authors should provide the pDOS of non-bonding oxygen states, which usually work as evidence for LOM. b), In the LOM pathway, there was no O-O coupling. The authors should reconduct the simulation.

Response:

We are grateful to the reviewer for the valuable suggestion. a) According to your suggestion, we have supplemented the pDOS of non-bonding oxygen states of CoO, Fe-CoO, F-CoO, and Fe, F-CoO slabs in the revised manuscript. For CoO and Fe-CoO, there is one kind of oxygen: O1, O-M-O (M=Co, Fe). For F-CoO and Fe, F-CoO, there are two kinds of oxygen: O1, O-M-O and O2, O-M-F (M=Co, Fe). As shown in **Supplementary Fig. 27**, large amounts of high-energy non-bonding oxygen states are generated, which are associated with the O2 bonding environment. The existence of non-bonding oxygen states after F doping is favorable for lattice oxygen oxidation (see related Refs. *Nat. Energy*, 2019, 4, 329; *Nat. Energy*, 2018, 3, 373). These results correspond well with the experiments and Gibbs free energy diagrams, demonstrating that F doping makes the LOM pathway more favorable. The related descriptions have been added in the revised manuscript (see Page 15).

Supplementary Fig. 27. PDOS of O 2p orbitals and non-bonding oxygen states (O2) in CoO, Fe-CoO, F-CoO and Fe, F-CoO slabs.

b) The AEM and LOM routes with five different elementary steps are schematically illustrated in **Figs. 5h and 5i**. For the LOM, the catalyst follows a single-metal-site mechanism, involving O-M-OH, M OO (dotted box), $\text{O}\text{O-M-}$, and M-OH (bold **O** (highlighted with red color) is lattice oxygen active site, \square denotes oxygen vacancy), which is consistent with the reported work (see related Ref. *Energy Environ. Sci.*, 2021, 14, 4647). Compared with AEM, the LOM differs in that it experiences the direct coupling of *O intermediate with active lattice oxygen (dotted box), resulting in the

generated O₂ molecule containing a lattice oxygen (see related Ref. Nat. Energy, 2019, 4,0329). The free energy diagrams reveal that the OER mechanisms of the CoO and Fe-CoO slabs both follow a conventional AEM route, since the potential determining step (PDS) in an AEM route, that is Co-OO formation, holds lower energy compared to oxygen vacancy formation in a LOM pathway (Fig. 5j and Supplementary Figs. 28, 29). Additionally, the energy barrier of the PDS for the AEM pathway decreases from 2.01 to 1.38 eV after Fe doping (Supplementary Table 7). This is in great alignment with the result that Fe-doping causes a positive shift of ϵ_{d} of active Co sites, thus enhancing the OER process. However, once F is doped into CoO, as seen from F-CoO and Fe, F-CoO slabs, the free energy change for the formation of oxygen vacancy in the LOM pathway decreases dramatically, implying that F doping plays a critical role in stabilizing the oxygen vacancy (Fig. 5k and Supplementary Figs. 30, 31). As a result, the LOM pathway is thermodynamically more favorable for F-CoO and Fe, F-CoO slabs with a lower energy barrier compared to the AEM pathway, further verifying the effective activation of lattice oxygen in F-CoO and Fe, F-CoO slabs. What's more, the Fe, F-CoO slab delivers the lowest energy uphill of 1.17 eV to complete the OER loop among the four samples, thereby exhibiting supreme intrinsic activity and a decreased overpotential for OER. These results consist well with the experimental phenomena.

Figure 5 Schematic illustrations of the OER mechanisms: (h) AEM and (i) LOM. (j) The Gibbs free energy diagrams of OER steps on the Fe, F-CoO, F-CoO, Fe-CoO, and CoO slabs via the AEM (h) and LOM (i) pathways. (k) Models of the different intermediates in the LOM pathway on the Fe, F-CoO slab.

Supplementary Fig. 28. Models of the different intermediates in AEM and LOM pathways on the CoO slab.

Supplementary Fig. 29. Models of the different intermediates in AEM and LOM pathways on the Fe-CoO slab.

Supplementary Fig. 30. Models of the different intermediates in AEM and LOM pathways on the F-CoO slab.

Supplementary Fig. 31. Models of the different intermediates in AEM pathways on the Fe, F-CoO slab.

Supplementary Table 7. Energy barriers of the PDS in LOM and AEM pathways over Fe, F-CoO NNAs, F-CoO NNAs, Fe-CoO NNAs, and CoO NNAs.

Catalysts	Energy barrier of the PDS in LOM (eV)	Energy barrier of the PDS in AEM (eV)
Fe, F-CoO	1.17	1.22
F-CoO	1.52	1.53
Fe-CoO	2.50	1.38
CoO	2.72	2.01

(5) The resolution of the figure. The authors should provide high-resolution figure.

Response:

We are grateful to the reviewer for the valuable suggestion. High-resolution figures have been provided in the revised manuscript and Supplementary Information.

Reviewer #3: Promotion of water oxidation remains a challenge that must be addressed for renewables to meet a major fraction of global energy demand. This study proposes a strategy which utilizes electric-field and Fe/F doping to promote water oxidation in the context of CoO electrocatalyst.

(1) The authors did not elaborate much on some aspects of their computational modeling. For example, there is no mention of the deployed DFT code and the flavor of DFT electrochemistry-based technique. Is the latter based on the Norskov's computational hydrogen electrode technique?

Response:

We are grateful to the reviewer for the valuable suggestion. In this work, the CoO (200) plane was chosen as an active interface, and the corresponding slabs of Fe, F-CoO, F-CoO, Fe-CoO and CoO were established based on $2 \times 2 \times 2$ supercell of CoO crystal structure, respectively. To avoid periodic interaction, a vacuum layer of 30 Å was incorporated into the slabs. The computations were conducted based on the density

functional theory (DFT) using Dmol3 and CASTEP code. Dmol3 code was utilized to optimize the structure of slabs and calculate the free energies, CASTEP code was employed for partial density of states (PDOS) of these optimal slabs. DFT calculations of these slabs were computed by using a generalized gradient approximation (GGA) of exchange-correlation functional in the Perdew, Burke, and Ernzerhof (PBE) (see related Ref. Phys. Rev. Lett., 1996, 77, 3865). The structure was fully optimized until the force on each atom is less than 10^{-3} Ha/Å. Moreover, to investigate the effect of local electric field on the OER process, an electric field of 1.58×10^{-2} V/ Å based on the finite element analysis numerical simulations, was introduced into the calculation of free energy on the Fe, F-CoO slab. PDOS calculation was performed by GGA+U functional with additional Coulomb potential ($U_{Fe} = 3.0$ and $U_{Co} = 3.1$ eV) for 3d-orbit, a plane-wave energy cut off of 500 eV was used together with norm-conserving pseudopotentials, and the Brillouin zone was sampled with a $2 \times 2 \times 1$ Monkhorst–Pack grid (see related Ref. Phys. Rev. B, 1976, 13, 5188).

$$\Delta G = \Delta E + ZPE - T\Delta S + ne\Delta U$$

where ΔE is the total energy, ZPE is the zero-point energy, the entropy (ΔS) of each adsorbed state is yielded from DFT calculation and ΔU is applied potential, whereas the thermodynamic corrections for gas molecules are from standard tables.

In this work, Norskov’s computational hydrogen electrode was applied to calculate the reaction ΔG for OER (see related Ref. Energy Technol., 2022, 10, 2200085). In the method, with the standard conditions ($pH = 0$, $p = 1$ bar, $T = 298$ K), the ΔG of the reaction:

could be calculated from the reactions:

$$\Delta G = G(AH^*) - G(1/2H_2) - G(A^*) + e\Delta U$$

The related calculation details have been provided in the revised Supplementary Information (see Pages S2 and S3).

(2) It appears that an extended low index facet model is used here to model reactions at the surfaces of the CoO nanostructures. While the use of such flat model should not be taken for granted, it does not account for the high curvature structures that may be

present on the active site. The authors themselves mention that it is the high-curvature feature that can effectively promote efficiency and improve current density in different electrocatalytic reactions (p. 4). In particular, Fe doping is mentioned to be highly influential in the generation of such morphology (p. 7). If CoO possesses a fcc-bulk structure, why not consider using the (211) slab to model a high curvature active site.

Response:

We are grateful to the reviewer for the insightful suggestion. We chose the CoO (200) slab rather than the (211) slab for theoretical calculations for three reasons. First, previous researches on high curvature structures often choose stable planes for theoretical calculations (see related Refs. *Sci. Adv.* 2022, 8, eabm9477; *Nano Lett.* 2022, 22, 1963; *Angew. Chem. Int. Ed.* 2020, 132, 8706). The CoO (200) facet is more stable compared to the CoO (211) crystal plane, and the (200) plane of CoO is usually chosen for theoretical calculations in most works (see related Refs. *Angew. Chem. Int. Ed.* 2020, 59, 6929; *Chem. Eng. J.*, 2021, 428, 131031; *Adv. Funct. Mater.* 2022, 32, 2109336; *Adv. Sci.* 2023, 10, 2300122). Second, the HRTEM image and SAED patterns show that the Fe, F-CoO NNAs catalyst exposes many active CoO (200) facets. However, the CoO (211) crystal plane is absent from the HRTEM, SAED and XRD patterns. Finally, to aid in interface selection, doping energies of Fe and F in CoO (200) and (211) were calculated in this work (Fig. R2). The doping energies (ΔG_{doping}) were computed from Eqns. R1- R4. The doping energy difference (ΔE_{doping}) is defined as the value subtracting ΔG_{doping} for the (211) slab from ΔG_{doping} for the (200) slab (Eqn. R5). A negative ΔE_{doping} indicates that the dopant prefers to doping into the (200) slab, while the verse suggests that it tends to dope in the (211) slab. As Fig. R3 illustrates, negative ΔE_{doping} are obtained for Fe and F dopants, indicating the Fe and F dopants tend to be doped into the CoO (200) rather than the CoO (211) slab. Considering the above, the (200) crystal plane of CoO was thus chosen as the theoretical model.

For Fe doping:

$$\Delta G_{doping-(200)} = (\Delta G_{Fe-doping-(200)} + \Delta G_{Co}) - (\Delta G_{pure-(200)} + \Delta G_{Fe}) \quad (R1)$$

$$\Delta G_{doping-(211)} = (\Delta G_{Fe-doping-(211)} + \Delta G_{Co}) - (\Delta G_{pure-(211)} + \Delta G_{Fe}) \quad (R2)$$

For F doping:

$$\Delta G_{doping-(200)} = (\Delta G_{F-doping-(200)} + \Delta G_{O}) - (\Delta G_{pure-(200)} + \Delta G_{F}) \quad (R3)$$

$$\Delta G_{doping-(211)} = (\Delta G_{F-doping-(211)} + \Delta G_{O}) - (\Delta G_{pure-(211)} + \Delta G_{F}) \quad (R4)$$

$$\Delta E_{doping} = \Delta G_{doping-(200)} - \Delta G_{doping-(211)} \quad (R5)$$

where ΔG_{Co} is the energy of single Co atom, and similarly, ΔG_{Fe} , ΔG_O and ΔG_F are the energies of single Fe, O and F atom, respectively.

Figure R2 Slab models of (a) CoO (200), (b) Fe-CoO (200), (c) F-CoO (200), (d) CoO (211), (e) Fe-CoO (211), (f) F-CoO (211).

Figure R3 ΔE_{doping} for Fe and F doping.

(3) The models for the doped CoO contain a single Fe and/or F atom situated on the top site. Is there any evidence that points to preferential dopant incorporation in the near surface region. While XRD pattern supports the incorporation of Fe and F in the CoO lattice, it does not approximately pinpoint where the dopants are in the nanoneedles. In the absence of experimental confirmation, maybe DFT thermodynamics analysis can help sort this out (see Phys. Chem. Chem. Phys, 21, 23626-23637 (2019)).

Response:

We are grateful to the reviewer for the valuable suggestion. According to your suggestion, DFT thermodynamics analysis was conducted on CoO to investigate the preferential positions of Fe and F dopants. In order to model the segregation, we considered a Fe/F dopant atom in the first (top) surface layer and a sub-layer (bulk)

(Supplementary Fig. 26). The segregation energy, E_{seg} , is defined as the total energy difference of the CoO slab with dopant in the first layer and sub-layer (Eqn. R6), based on the previous report (see related Ref. Phys. Chem. Chem. Phys, 2019, 21, 23626). The E_{seg} of Fe in CoO (200) surface is -1.94 eV, indicating that Fe dopant prefers to segregate towards the surface. While the E_{seg} of F in CoO (200) surface is 0.79 eV. The relatively small E_{seg} value suggests that the trend for doping F into the bulk and on the surface is similar. These theoretical results are consistent with the experiments. As shown by XPS, 9.8 at.% of Fe and 6.0 at.% F are observed on the surface, while only 0.6 at.% of Fe is found with EDS mapping, verifying that almost all the Fe atoms are incorporated on the surface of CoO. More importantly, it is believed that electrochemical reactions occurred at the surface and allow surface formation of O–O bonds because of the low bulk flexibility (see related Ref. Nat. Mater., 2016, 15, 121) Considering the above, the CoO models with Fe and F atoms situated on the top site were constructed to investigate the effect of Fe and F dual dopants on the surface lattice oxygen. We have added these results in the revised manuscript (see Page 14).

$$E_{seg} = E_{1st-layer} - E_{sub-layer} \quad (R6)$$

where E_{seg} is the calculated segregation energy, $E_{1st-layer}$ is the total energy of the alloy with the dopant in the first layer, and $E_{sub-layer}$ is the total energy of the alloy with the dopant in the sub-layer.

Supplementary Fig. 26 Slab models for (a) F dopant within the first layer of CoO, (b) F dopant within the sub-layer of CoO, (c) Fe dopant within the first layer of CoO, and (d) Fe dopant within the sub-layer of CoO.

(4) A particular highlight of this work is high curvature of the CoO nanoneedles results in a large local electric field on their tips which can promote the reaction. In this work, finite element simulation method is used to explore this further. Another way to look at

the role of the electric field is to use DFT. The authors must have used VASP in this work, a software that allows reaction modeling in the presence of electric field.

Response:

We are grateful to the reviewer for the valuable suggestion. Based on the finite element analysis numerical simulations, the tips of rough nanoneedle can display a large positive charge density of 0.11 C m^{-2} , corresponding to an electric field of $1.58 \times 10^{-2} \text{ V/\AA}$. Therefore, to explore the effect of the local electric field on the OER process, an electric field of $1.58 \times 10^{-2} \text{ V/\AA}$ is introduced into the calculation of free energy on Fe, F-CoO slab. As illustrated in Supplementary Fig. 34, at an electric field of $1.58 \times 10^{-2} \text{ V/\AA}$, the energy uphill of PDS on Fe, F-CoO slab is lowered to 0.99 eV, indicating an enhanced OER activity.

Supplementary Fig. 34. The Gibbs free energy diagrams of OER steps via LOM pathway on Fe, F-CoO slab without and with an electric field ($1.58 \times 10^{-2} \text{ V/\AA}$).

At last, we wish to thank the Reviewers and the Editor again for the very constructive comments and suggestions to improve the quality of our manuscript. Thank you very much!

Reviewers' comments:

Reviewer #1 (Remarks to the Author):

Even though the authors have taken efforts to made revisions , but the overall level of innovation is still insufficient. It is not sufficiently appealing in terms of new material preparation and mechanistic interpretation. So this work is not suitable to be published in Nature Communications.

1. The catalyst preparation method presented in the manuscript is not innovative, as similar catalyst morphologies and preparation routes have been reported in literature (DOI: <https://doi.org/10.1002/chem.201904352>). There are also works on F-doping promoting lattice oxygen (DOI: <https://doi.org/10.1002/anie.202301408>) and the effect of local electric fields on OER (DOI: <https://doi.org/10.1002/adma.202007377>). Despite the extensive characterization efforts, the manuscript lacks a sufficiently compelling innovative aspect, making it difficult to quickly discern the highlight of the work: whether it lies in mechanism elucidation or innovative catalyst preparation? It appears that neither aspect is at the state-of-the-art level.
2. The manuscript lacks clear focus, it involves with a mix of various concepts, including lattice oxygen, local electric fields, doping, nanomorphology, and more. The entire work appears to be a compilation of diverse viewpoints, lacking a well-defined central theme. The connection between lattice oxygen promotion, electric field enhancement, and element doping is not clearly elucidated. The objectives and results are not well-defined, and the research question is not clearly articulated.
3. The quality of the images remains subpar, with incomplete text in Figure 1a and Figures 3ab, and many images lacking sufficient clarity.

Reviewer #2 (Remarks to the Author):

The authors addressed the questions well. No more comments.

Reviewer #3 (Remarks to the Author):

The authors have addressed all the issues/corrections I pointed out in my previous report.

Response to Reviewers:

We thank the reviewers for their time and very useful comments in improving the quality of this manuscript. Provided below is our detailed response to each question.

Reviewers' comments:

Reviewer #1: Even though the authors have taken efforts to made revisions, but the overall level of innovation is still insufficient. It is not sufficiently appealing in terms of new material preparation and mechanistic interpretation. So this work is not suitable to be published in Nature Communications.

1. The catalyst preparation method presented in the manuscript is not innovative, as similar catalyst morphologies and preparation routes have been reported in literature (DOI: <https://doi.org/10.1002/chem.201904352>). There are also works on F-doping promoting lattice oxygen (DOI: <https://doi.org/10.1002/anie.202301408>) and the effect of local electric fields on OER (DOI: <https://doi.org/10.1002/adma.202007377>). Despite the extensive characterization efforts, the manuscript lacks a sufficiently compelling innovative aspect, making it difficult to quickly discern the highlight of the work: whether it lies in mechanism elucidation or innovative catalyst preparation? It appears that neither aspect is at the state-of-the-art level.

Response: We'd like to thank the review for the valuable comments, and we are very grateful to the reviewer for providing the important references. We have thoroughly studied the works that the reviewer kindly provided. For instance, similar morphology effect to our work has been reported (DOI: [10.1002/chem.201904352](https://doi.org/10.1002/chem.201904352)), and the lattice oxygen effect and the local electric field effect have been adopted to promote OER, respectively, in those works (DOI: [10.1002/anie.202301408](https://doi.org/10.1002/anie.202301408), DOI: [10.1002/adma.202007377](https://doi.org/10.1002/adma.202007377)). However, every reported work deals only with a single effect. **In sharp contrast, we have integrated both electrode reaction and mass transfer in one catalyst material. To our knowledge, this integration of two effects has never been studied yet.**

Based on the size compatibility of Fe and Co, F and O, and the large electronegativity of F, the simple strategy of dual doping of cation and anion is thus developed for simultaneously realizing lattice oxygen activation and local electric field

enhancement for OER in our work. More importantly, to clarify the synthetic effect of lattice oxygen activation and local electric field on OER and their relationship with Fe and F co-doping, X-ray absorption spectroscopy, in situ Raman spectroscopy, chemical probe, ^{18}O isotope technology, differential electrochemical mass spectrometry, density functional theory calculations, finite element method simulations, and in situ optical microscope were conducted. Both experimental measurements and theoretical simulations have disclosed that **dually doped Fe and F cooperatively tailor the electronic properties of CoO, and lead to the improved covalency of the metal-oxygen band and stimulated lattice oxygen. Meanwhile, the doping of Fe not only induces a synergetic effect of tip enhancement and proximity effect, but also enlarges the local electric field to effectively concentrate OH^- ions, which consequently improves the OER kinetics by optimizing the reaction energy barrier and promoting O_2 desorption.** Benefiting from the synergetic effect of lattice oxygen activation and local electric field induced by co-doping of Fe and F, the thermodynamics and kinetics for OER are promoted simultaneously, leading to boosted electrocatalytic performance. To specify the innovation of the paper, we have improved the manuscript as in **Pages 2, 5 and 19.**

2. The manuscript lacks clear focus, it involves with a mix of various concepts, including lattice oxygen, local electric fields, doping, nanomorphology, and more. The entire work appears to be a compilation of diverse viewpoints, lacking a well-defined central theme. The connection between lattice oxygen promotion, electric field enhancement, and element doping is not clearly elucidated. The objectives and results are not well-defined, and the research question is not clearly articulated.

Response: We are grateful to the reviewer for the valuable comment. Our work is mainly focused on the conceptual strategy of coupling lattice oxygen and local electric field effects in the field of OER. To simultaneously realizing active lattice oxygen and enhanced local electric field, the method of dual doping of cation and anion have been developed. To further elucidate connection between lattice oxygen promotion, electric field enhancement, and co-doping of Fe and F, X-ray absorption spectroscopy, in situ Raman spectroscopy, chemical probe, ^{18}O isotope technology, differential electrochemical mass spectrometry, density functional theory calculations, finite element method simulations, and in situ optical microscope were conducted. Both experimental measurements and theoretical simulations have disclosed that dually

doped Fe and F cooperatively tailor the electronic properties of CoO, and lead to the improved covalency of the metal-oxygen band and stimulated lattice oxygen. Besides, the Fe and F dopants mainly adjust the electronic states of the Co atom and oxygen ligand, respectively, leading to the disparate electrochemical behaviors and OER mechanisms in single-doped simples. Meanwhile, it is confirmed that the Fe doping induces a synergetic effect of tip enhancement and proximity effect contributing to a stronger electric field, which favors the enrichment of OH⁻ around the active sites, optimizes the energy barrier for lattice oxygen oxidation process, and promotes the release of O₂ bubbles. Benefiting from the synthetic effect of lattice oxygen activation and local electric field, the OER are boosted significantly.

In addition, we have revised the related descriptions including title and abstract in the manuscript to clarify the focus and the connection between lattice oxygen promotion, electric field enhancement, and element doping (Pages 2, 5, 19). This integration concept of lattice oxygen activation and local electric field enhancement is novel and efficient for industry water oxidation (see Figure R1).

Figure R1 The connection between Fe and F co-doping, lattice oxygen promotion, and electric field enhancement. This integration concept of lattice oxygen activation and local electric field enhancement is innovative.

3. The quality of the images remains subpar, with incomplete text in Figure 1a and Figures 3ab, and many images lacking sufficient clarity.

Response: We are grateful to the reviewer for pointing this out. We have improved the

quality of the images with high-resolution of 2400 dpi and the related descriptions in the text.

Reviewer #2: The authors addressed the questions well. No more comments.

Response: We appreciate the reviewer for reorganization of our work and we have further improved the quality of the manuscript.

Reviewer #3: The authors have addressed all the issues/corrections I pointed out in my previous report.

Response: We appreciate the reviewer for reorganization of our work and we have further improved the quality of the manuscript.

At last, we wish to thank the Reviewers and the Editor again for the very constructive comments and suggestions to improve the quality of our manuscript. Thank you very much!

REVIEWERS' COMMENTS

Reviewer #1 (Remarks to the Author):

While the manuscript shows an in-depth investigation and mechanistic interpretation of Fe-F co-doped CoO nanoneedle array electrode, we still find the innovativeness of such research insufficient for publication in Nature Communications.

1. The manuscript describes the induced lattice oxygen activation and local electric fields effects, which is intriguing. However, both these mechanisms have been previously reported. The manuscript fails to demonstrate synergistic effects or mutual influence between the local electric fields and Fe-F co-doping, merely superimposing these two mechanisms, which does not substantiate its innovativeness.
2. While the authors introduce these two mechanisms, their reliance on existing theories results in a lack of a deeper understanding of the relationship between catalytic activity and structure, as well as the catalytic mechanisms. Furthermore, the material synthesis and preparation methods lack novelty. In summary, we believe this work is not suitable to be published in Nature Communications.

Dec. 27, 2023

Manuscript ID: NCOMMS-23-30719B-Z

Title: Lattice oxygen activation and local electric field enhancement by co-doping Fe and F in CoO nanoneedle arrays for industrial electrocatalytic water oxidation

We thank reviewer for precious time on our manuscript and very constructive comments in improving the quality of this manuscript. Provided below is our detailed response to each question.

Reviewer's comments:

Reviewer #1: While the manuscript shows an in-depth investigation and mechanistic interpretation of Fe-F co-doped CoO nanoneedle array electrode, we still find the innovativeness of such research insufficient for publication in Nature Communications. (1) The manuscript describes the induced lattice oxygen activation and local electric fields effects, which is intriguing. However, both these mechanisms have been previously reported. The manuscript fails to demonstrate synergistic effects or mutual influence between the local electric fields and Fe-F co-doping, merely superimposing these two mechanisms, which does not substantiate its innovativeness.

Response: We'd like to thank the reviewer for the valuable comments. Previous reported studies just achieved only one effect that contributes to improved electrocatalytic performance. It's really challenging to achieve both lattice oxygen and local electric field effects at the same time, since the reaction rate is highly related to the electrode reaction and mass transfer to the electrode surface. **The innovation of our work is just that we have realized the integration of lattice oxygen and local electric field effects by only the dual doping of cation and anion in one catalyst material, which has never been reported before.** Moreover, by combining X-ray absorption spectroscopy, in situ Raman spectroscopy, chemical probe, ¹⁸O isotope technology, differential electrochemical mass spectrometry, density functional theory (DFT) calculations, finite element method simulations, and in situ optical microscope, the synthetic effect of lattice oxygen activation and local electric field on OER and their relationship with Fe and F co-doping (Figure R1) have been established. Both experimental measurements and theoretical simulations have disclosed that **dually doped Fe and F cooperatively tailor the electronic properties of CoO, and lead to**

the improved covalency of the metal-oxygen band and stimulated lattice oxygen. Meanwhile, the doping of Fe induces a synergetic effect of tip enhancement and proximity effect, significantly enlarging the local electric field. The high local electric field is favorable for concentrating more OH⁻ ions, promoting O₂ desorption, and lowering the energy barrier for lattice oxygen activation. These results verify that the strong local electric field can effectively promote the kinetics of lattice oxygen oxidation mechanism (LOM), which had not been shown before. The related descriptions are shown in the manuscript (Pages 2, 5, 19).

Figure R1 The connection between Fe and F co-doping, lattice oxygen promotion, and electric field enhancement. This integration concept of lattice oxygen activation and local electric field enhancement is innovative.

(2) While the authors introduce these two mechanisms, their reliance on existing theories results in a lack of a deeper understanding of the relationship between catalytic activity and structure, as well as the catalytic mechanisms. Furthermore, the material synthesis and preparation methods lack novelty. In summary, we believe this work is not suitable to be published in Nature Communications.

Response: We'd like to thank the reviewer for the valuable comments. To deeper understand the relationship between catalytic activity, structure, and catalytic mechanisms, the influences of Fe and F dual doping on electronic structure, morphology, and OER performance, as well as the enhancing mechanism and underlying OER mechanism are systematically investigated by combining X-ray absorption spectroscopy, in situ Raman spectroscopy, chemical probe, ¹⁸O isotope

technology, differential electrochemical mass spectrometry, density functional theory calculations, finite element method simulations, and in situ optical microscope. Both experimental measurements and theoretical simulations have disclosed that dually doped Fe and F cooperatively tailor the electronic properties of CoO, and lead to the improved covalency of the metal-oxygen band and stimulated lattice oxygen. Besides, the Fe and F dopants mainly adjust the electronic states of the Co atom and oxygen ligand, respectively, leading to disparate electrochemical behaviors and OER mechanisms in single-doped samples. Meanwhile, the doping of Fe introduces a rough surface, which not only induces a synergetic effect of tip enhancement and proximity effect, but also enlarges the local electric field to effectively concentrate OH⁻ ions and improve the OER kinetics by optimizing the energy barrier of LOM pathway and promoting O₂ desorption. Benefiting from the synergetic effect of lattice oxygen activation and local electric field, Fe, F-CoO NNAs exhibit superior electrocatalytic performance.

To our knowledge, this integration of lattice oxygen and local electric field effects has never been studied yet. And the strategy of cation and anion dual doping and the catalyst of Fe and F co-doped CoO NNAs to realize lattice oxygen activation and local electric field enhancement simultaneously for OER is for the first time proposed.

At last, we wish to thank the Reviewers and the Editor again for the very constructive comments and suggestions to improve the quality of our manuscript.